# VOLTA: an enVironment-aware cOntrastive ceLl represenTation leArning for histopathology

Ramin Nakhli [1,9], Katherine Rich[2,9], Allen Zhang[3], Amirali Darbandsari[4], Elahe Shenasa[3], Amir Hadjifaradji[1], Sidney Thiessen[5], Katy Milne [5], Steven J. M. Jones [6,7], Jessica N. McAlpine [8], Brad H. Nelson[5], C. Blake Gilks[3], Hossein Farahani [1,10] & Ali Bashashati [1,3,6,10] ✉

In clinical oncology, many diagnostic tasks rely on the identification of cells in histopathology images. While supervised machine learning techniques necessitate the need for labels, providing manual cell annotations is time-consuming. In this paper, we propose a self-supervised framework (enVironment-aware cOntrastive cell represenTation learning: VOLTA) for cell representation learning in histopathology images using a technique that accounts for the cell's mutual relationship with its environment. We subject our model to extensive experiments on data collected from multiple institutions comprising over 800,000 cells and six cancer types. To showcase the potential of our proposed framework, we apply VOLTA to ovarian and endometrial cancers and demonstrate that our cell representations can be utilized to identify the known histotypes of ovarian cancer and provide insights that link histopathology and molecular subtypes of endometrial cancer. Unlike supervised models, we provide a framework that can empower discoveries without any annotation data, even in situations where sample sizes are limited.

Cells located within the micro-environment of a tumor have a prominent impact on its developmental process[1–5]. Variations in the micro-environment have been associated with the epigenetic profiles within the tumor and the heterogeneity in the associated gene expression profiles[6]. Various cell types reside in the tumor micro-environment and growing evidence suggest that intratumoral heterogeneity is a large contributing factor to the therapeutic resistance of the tumor[6,7]. Several studies have shown that higher levels of intratumoral heterogeneity are strongly associated with poor outcomes in lung, ovarian, head and neck, and pancreatic cancers, with implications that the tumor is more likely to harbor a rare pre-existing resistant subclone[6,8–10]. Furthermore, spatial distribution of immune cells within the tumor microenvironment has a significant impact on the prognosis and therapeutic responses[4,11–14]. Therefore, the identification of individual cells within the tumor micro-environment is a vital step for tumor characterization in many complex tasks such as tissue classification, cancer diagnosis, subtyping and histological grading[15–18].

The visual assessment of the Hematoxylin & Eosin (H&E)-stained tissue slides under the microscope is the conventional and widely utilized approach to tumor characterization and cell identification. However, manual cell identification can be cumbersome due to the time-consuming nature of the assessment of large numbers of cells

[1]School of Biomedical Engineering, University of British Columbia, Vancouver, BC, Canada. [2]Bioinformatics Graduate Program, University of British Columbia, Vancouver, Canada. [3]Department of Pathology and Laboratory Medicine, University of British Columbia, Vancouver, BC, Canada. [4]Department of Electrical and Computer Engineering, University of British Columbia, Vancouver, BC, Canada. [5]Deeley Research Centre, BC Cancer Agency, Victoria, BC, Canada. [6]Canada's Michael Smith Genome Sciences Centre, BC Cancer Research Institute, Vancouver, Canada. [7]Department of Medical Genetics, University of British Columbia, Vancouver, Canada. [8]Department of Obstetrics and Gynecology, University of British Columbia, Vancouver, BC, Canada. [9]These authors contributed equally: Ramin Nakhli, Katherine Rich. [10]These authors jointly supervised this work: Hossein Farahani, Ali Bashashati. ✉e-mail: ali.bashashati@ubc.ca

(tens of thousands in a single slide) and suffers from pathologists' intra- and inter-observer variability[19]. Machine learning and deep learning models coupled with the digitization of pathological material offer opportunities for computer-aided cell identification[20–22]. Despite the long history of machine learning research in cell classification using handcrafted features[23–25], significant improvements have been reported by employing deep learning-based models[21]. For example, in a recent study[26], authors developed a pipeline for segmentation and identification of several molecular features of cells from H&E images by employing supervised techniques while the ground truth data (i.e., labels) were generated through immunohistochemistry (IHC) staining and co-registration of IHC and H&E images.

Even though supervised models can potentially reduce the manual workload of cell identification, they require a large number of cell-level annotations for training. However, generating annotations requires labor-intensive manual examination of the tissue by pathologists. Furthermore, a model trained on a specific tissue type (e.g., ovarian cancer) cannot be directly applied to another tissue type (e.g., breast cancer); therefore, the data collection and labeling process has to be carried out again to retrained the model for a new tissue type. To address this issue, several studies have utilized unsupervised approaches for cell representation learning and clustering[27]. adopt InfoGAN[28] to train an implicit classifier, and in another attempt[29], use a deep convolutional auto-encoder (DCAE) to learn the embeddings of cells. However, these studies focus on a single tissue type, which may not generalize to other tissues. Additionally, these techniques ignore the surrounding environment of a cell. Many recent studies have shown that cells are directly impacted by their environment[30–32] and as such, incorporation of the environment information may improve the performance of the models.

Recently, self-supervised learning (SSL) techniques have emerged as an important step towards generalizable representation learning. SSL is a technique developed for image representation learning, guided by using the augmentations of an image as its label. The utility of this technique has been investigated on different tasks in the natural image domain where[33] demonstrate the capability of this technique in object classification, and[34] show its efficacy in object detection. Despite the fact that a few studies[35,36] examine the utility of self-supervised methods in the patch-level classification of histopathology images, the potential of self-supervised techniques for labeling individual cells (rather than just classifying image patches) are largely ignored. More importantly, cell-based representation and classification techniques provide better linkages to biological mechanisms and tumor micro-environment assessment while patch-based techniques may fail to provide more explainable linkages to biology.

In this work, we propose a self-supervised framework for cell representation learning in histopathology images by introducing a technique to incorporate the mutual relationship between the cell and its environment for improved cell representation. We benchmark our model on data representing more than 800,000 cells in four cancer histotypes with three to six cell types in each dataset. Results confirm the superiority of our model in memory-efficient cell type representation compared to the state-of-the-art. We further utilize the proposed model in the context of ovarian and endometrial cancers and demonstrate that our cell representations, without any human annotations, can be utilized to identify the known histotypes of ovarian cancer, and gain novel insights that link histopathology and molecular subtypes of endometrial cancer.

## Results
### Cell representation learning framework and benchmarking
Figure 1 depicts an overview of our proposed enVironment-aware cOntrastive cell represenTation leArning model (VOLTA). This framework consists of two major blocks, *Cell Block* and *Environment Block*. The *Cell Block* takes an image of a cell and applies two sets of augmentation operations to create visually distinct perspectives of the

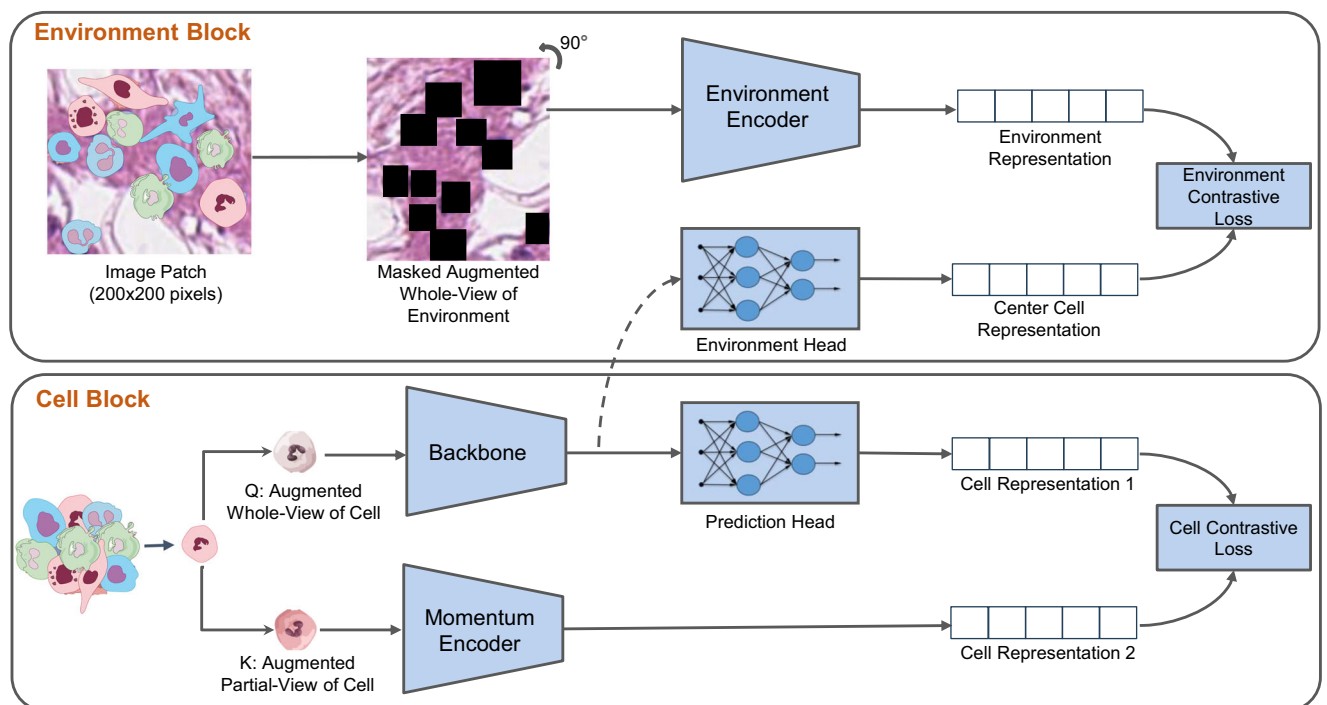

**Fig. 1 | Overview of our proposed framework.** Overview of our proposed framework. The *cell block* trains the backbone model by applying two augmentations on the same cell image, encoding the images, and bringing their representations close to each other. While the backbone is trained through back-propagation, the momentum encoder averages the weights from the backbone. On the other hand, the *Environment Block* combines the cell representation created by the *cell block* with the surrounding environment (a larger region around the cell). We mask all of the cells in the environment patch to prevent the model from favoring the cell representation toward that of these cells (Source data are provided as a Source Data file).

cell. This structure is inspired by the architectural design of self-supervised models[37,38]. The main purpose of doing so is to have two visually different-looking images of the exact same cell. These two augmented images are then transformed into their respective representation vectors using a stack of deep neural networks and, given that these representations correspond to the same cell, the models are trained to minimize the distance between the two representations. Even though it is possible to utilize more than two branches (i.e., more than two sets of augmentations), the two-branch design prevents complications in the pipeline and the loss function.

The *Environment Block* of our proposed framework is utilized to increase the mutual information between the cell and a larger patch that captures the environment surrounding it. Specifically, we hypothesize that there is a mutual information between each cell and its environment; therefore, we aim to maximize this mutual information during training. By using the InfoNCE loss function[39], VOLTA accomplishes this by performing a contrastive cross-modal learning between the cell representation and that of its environment. To prevent the model from biasing towards other cells appearing in the environment, we mask out these cells in the environment patch before feeding it to the model. Finally, the cell representations for downstream tasks such as cell clustering and classification can be obtained by using the backbone model trained in this setting.

We benchmarked these representations across multiple tasks and datasets. More specifically, nine public and private datasets (CoNSeP[21], NuCLS[22], Pannuke Breast[40], Pannuke Colon[40], Lizard[41], SarcCell, Oracle, MastCell, and MiDOG[42]) representing 800,000 cells and six cancer types (colon, breast, and ovarian, skin, neuroendocrine, and sarcoma) were utilized to evaluate the performance of the proposed cell representation model (Supplementary Note 1 and Supplementary Table 1). Even though our model requires no labels for training, we split the data into train and test sets and use the former for the training of the model.

We also conducted ablation studies on the separate components of our model to measure their effects on the performance (see Supplementary Note 2). Our experiments suggest that the cell masking operation (Supplementary Table 2), whole- and local-view augmentations (Supplementary Tables 3 and 4), memory storage (Supplementary Table 5), environment patch size (Supplementary Table 6), and momentum encoder (Supplementary Table 7) provide noticeable performance improvements to our model.

## Identification of distinct cell clusters by self-supervised cell representation learning

VOLTA produces cell representations from histopathology images, and these representations should be capable of differentiating between biologically distinct cell types. To test this hypothesis, we used our method to identify cell clusters in each dataset. To be specific, after learning the cell representations in a self-supervised manner using VOLTA, we performed unsupervised clustering on the cell representations and examined the enrichment of the identified clusters with specific cell types. To show the utility of our approach, we compared the performance of VOLTA with the state-of-the-art morphology-based and deep learning-based models for cell representation. As shown in Table 1, our model outperformed all counterparts by a large margin across multiple clustering metrics in all datasets (adjusted mutual index (AMI)[43], adjusted rand index (ARI)[44], Purity[45], Dunn Index, and Silhouette Score - see Supplementary Note 3, Supplementary Note 4, and Supplementary Table 8), reaching twice the performance of the best-performing baselines in some of the datasets (except for Oracle and SarcCell datasets where SimCLR and GAN perform better, respectively). More importantly, while the performance of the baseline models varies from one cancer to another, our model shows consistent results regardless of the cancer type. As an example, while the morphology-based representation method has the

best performance compared to the other baselines over the NuCLS and PanNuke Breast cancer datasets, it has an inferior performance on PanNuke Colon and CoNSeP.

Figure 2 and Supplementary Fig. 1 (Supplementary Note 5) show the Uniform Manifold Approximation and Projection (UMAP) representations of various cell types that were derived by VOLTA using a contour-based and point-based visualization, respectively. The learned representations provide distinct and separable cell populations, thus confirming the comparison metrics that were presented in Table 1. Additionally, one can observe that our model is able to differentiate between immune cells (T-cell and B-cell) and tumor cells in the Oracle dataset. While this behavior can be seen in the SimCLR baseline, it is not observed in the other baselines (Supplementary Figs. 2–4 and Supplementary Note 6). Similarly, in the NuCLS dataset, our model is able to differentiate between stromal tumor-infiltrating lymphocytes (sTILs) and cancer cells. The same observations can be seen in the PanNuke Colon and CoNSeP datasets where various cell types such as epithelial and inflammatory cells are mapped to distinct locations in the embedding space.

## Supervised cell classification accuracy and efficiency improvement

We then aimed to assess the effectiveness of the proposed model in few-shot cell classification in a supervised machine learning setting where labeled samples were available. Specifically, we trained the model using our self-supervised framework and utilized the learned cell representations as inputs for training a simple Multi-Layer Perceptron (MLP) for cell classification. The performance of the trained model on CoNSeP and NuCLS datasets across various settings is shown in Fig. 3.

We also demonstrated the effectiveness of our self-supervised cell representation learning framework by using a subset of the labeled cell identities to train an MLP-based cell classifier. Our results showed that the proposed model achieved a reasonable performance with a small subset of the labeled training data (Supplementary Table 9 and Supplementary Note 7). For instance, with only 0.1% of the training labels, our models achieved 62.7% and 72.6% Top-1 accuracy on the CoNSeP and NuCLS datasets, respectively, while a model that utilized the entire labeled dataset achieved 80.2% and 76.3%. Furthermore, as the number of training labels increased, the classification accuracy consistently improved to an extent that our model outperformed the state-of-the-art Hover-Net model[21] results on the CoNSeP dataset, even with 70% of the training data. It is of note to mention that the number of the parameters of our proposed model is reduced by 60% compared to the HoVer-Net model (Supplementary Table 10 and Supplementary Note 8). Our model reached an accuracy that was close to the Masked-RCNN model which led to state-of-the-art results in the NuCLS dataset[22].

## Self-supervised cell representation learning is robust to undesired color variations

Previous studies have shown that normalization and domain adaptation methods can enhance the performance of supervised models when the train and test datasets are collected from different sites[46]. Given that the training and validation sets of NuCLS dataset are collected from different sites, we hypothesize that variations in staining and color profiles could lead to over-fitting of the supervised models to the training data. Therefore, we studied the effect of such methods on our proposed model when it was utilized for cell representation learning and supervised cell classification settings. To serve this purpose, we used the Vahadane normalization method[47] within the context of the NuCLS dataset where the slides were stained and scanned in different institutions.

Supplementary Table 11 illustrates the effect of the normalization in the self-supervised setting on the NuCLS dataset. Although[46] showed

**Table 1 | Unsupervised clustering of cell representations across different methods and datasets**

| Model | Metric | CoNSeP | NuCLS | PanNuke Breast | PanNuke Colon | Lizard | Oracle | SarcCell | MastCell | MiDOG |
|---|---|---|---|---|---|---|---|---|---|---|
| Pre-trained ImageNet | AMI | 7.3% | 9.3% | 5.42% | 11.21% | 6.25% | 0.26% | 0.8% | 0.1 % | 13.1% |
| | ARI | 7% | 7.8% | 3.94% | 8.21% | 4.36% | 0.42% | 1.8% | 0.1 % | 5.8 % |
| | Purity | 42.7% | 56.7% | 41.15% | 43.93% | 50.4% | 48.87% | 42.0% | 58.1% | 62.1% |
| Morphological | AMI | 12.7% | 21.1% | 8.94% | 7.88% | 13.21% | – | – | 0.0 % | – |
| | ARI | 1.3% | 18.8% | 7.28% | 6.19% | 9.22% | – | – | 0.0 % | – |
| | Purity | 48.8% | 66.1% | 47.06% | 42.73% | 57.5% | – | – | 58.1% | – |
| Manual Features | AMI | 9.5% | 11.25% | – | 7.86% | 10.2% | 2.74% | 2.9% | 2.1 % | 6.1% |
| | ARI | 6.4% | 7.8% | – | 6.53% | 3.8% | 2.24% | 2.1% | 4.3 % | 7.4% |
| | Purity | 45.5% | 56.2% | – | 40.37% | 52.9% | 53.84% | 42.7% | 62.0% | 63.7% |
| DCAE | AMI | 10.1% | 8.3% | 6.41% | 11.43% | 4.36% | 3.93% | – | 0.0 % | 3.5% |
| | ARI | 7.3% | 7.2% | 5.11% | 10.01% | 2.34% | 3.84% | – | 0.0 % | 4.3% |
| | Purity | 50.5% | 56.8% | 43.49% | 45.18% | 49.38% | 58.69% | – | 58.1% | 60.5% |
| GAN | AMI | 14.8% | 14% | 6.7% | 13.7% | 7.5% | 4.1% | **6.0**% | 0 % | 21.4% |
| | ARI | 15.7% | 12.6% | 4.6% | 11.4% | 3% | 5.8% | 5.6% | 0 % | 27.9% |
| | Purity | 58.4% | 62% | 42.4% | 49.6% | 48.9% | 57.5% | **46.0**% | 58.0% | 76.5% |
| SimCLR | AMI | 19.6% | 20.1% | 10.7% | 13.9% | 16.5% | **12.5**% | 5.6% | 6.2% | 30.2% |
| | ARI | 16.7% | 22.1% | 8.6% | 8.9% | 11.1% | **14.2**% | 4.5% | 8.4% | 30.7% |
| | Purity | 57.5% | 68.2% | **48.3**% | 40.9% | 57.1% | **67.5**% | 45.2% | 65.1% | 77.7% |
| DINO | AMI | 1.9% | 0.6% | 0.3% | 7.5% | 0.4% | 0.4% | 0.3% | 0.0 % | 4.9% |
| | ARI | 1.7% | 0.7% | 0.6% | 5.5% | 0.0% | 0.5% | 0.8% | 0.5% | 6.6% |
| | Purity | 1.9% | 0.7% | 0.5% | 7.1% | 0.4% | 0.6% | 41.9% | 58.1% | 62.9% |
| VOLTA (w/o env) | AMI | 24.2% | 22.8% | 10.75% | 19.5% | 10.85% | 3.6% | 5.3% | 0 % | 35.5% |
| | ARI | **21.7**% | 24% | 7.58% | 16.1% | 6.2% | 2.45% | 3.7% | 0 % | 39.4% |
| | Purity | 51.3% | 68.3% | 46.87% | 54.6% | 52.66% | 54.7% | 43.8% | 58.1% | 81.4% |
| VOLTA | AMI | **25.5**% | **26.2**% | **13.8**% | **22.5**% | **17.3**% | 8.05% | 4.2% | **25.4**% | **50.4**% |
| | ARI | 19.3% | **27.3**% | **8.94**% | **21.8**% | **11.4**% | 4.95% | **6.7**% | **33.1**% | **60.3**% |
| | Purity | **63.5**% | **70.3**% | 47.7% | **56.9**% | **57.9**% | 59.45% | 44.8% | **79.0**% | **88.8**% |

The best performance is shown in bold.

The baseline models include both morphology-based and state-of-the-art deep learning methods for cell representation. Some of the baseline results are listed as "–" meaning calculation of the feature vectors was not possible due to the limitation of the model on the small-sized cells.

that patch and slide classification tasks can benefit from cross-institution stain normalization, we noticed that our self-supervised cell representation approach does not benefit much from color normalization strategies. This finding can be attributed to the strong augmentations that were utilized in our self-supervised model training. Moreover, we investigated the effect of color normalization in the supervised fine-tuning setting. Interestingly, although self-supervised clustering results were robust to stain normalization, the supervised fine-tuned model benefited from it to an extent that it outperformed the MaskRCNN model[22] on this dataset (Supplementary Table 12 and Supplementary Note 9). It is of note to mention that the normalization method was only applied to the test set while the self-supervised model was still trained on the original data (i.e., without any normalization).

**VOLTA as a building block for unsupervised cancer subtype identification**

We sought to investigate the utility of our proposed self-supervised cell representation model as a building block for annotation-free cancer subtyping. Therefore, we put together a TMA cohort of 12 ovarian cancer cases comprising of clear cell, endometrioid, high-grade serous, and low-grade serous ovarian carcinomas. Applying the same procedure as described in 2.1, we utilized the cells extracted from these images to train our self-supervised model. Subsequently, after applying VOLTA, we extracted cell cluster distributions for each of the TMA core images and used them to perform hierarchical clustering to group the patients (see Supplementary Fig. 5). The results demonstrate that our model is capable of separating the epithelial ovarian cancer histotypes without a need for annotation or prior knowledge of the histotypes (Fig. 4a). In particular, four major clusters enriched with each of the four specific histotypes were identified with only two cases that were grouped with other subtypes. These results suggest an 91% accuracy (11 out of 12 that were correctly grouped) in ovarian cancer subtyping; a finding that is in line with results reported in the literature[48].

We next visualized the identified cell clusters on multiple patches and combined the clusters with similar cell types as assessed by a pathologist. We observed that each of the cell clusters is typically enriched with a specific type of cell, demonstrating the capability of the model in capturing morphological differences between cell types (Supplementary Figs. 6–10). Supplementary Table 13 represents the cell distributions across the epithelial ovarian histotypes after combining the initial cell clusters, while Supplementary Fig. 11 depicts the boxplot of the cell distributions before combination. Notably, we observed that the five identified cell clusters represented variations in tumor cell morphology associated with ovarian cancer histotypes. High-grade serous and clear cell tumors were relatively enriched for tumor cell clusters containing larger cells (tumor clusters 2, 4, and 5) compared to low-grade serous and endometrioid tumors (see Supplementary Figs. 12 and 13), consistent with the well-known high-grade nuclear histology of high-grade serous and clear cell carcinomas[49].

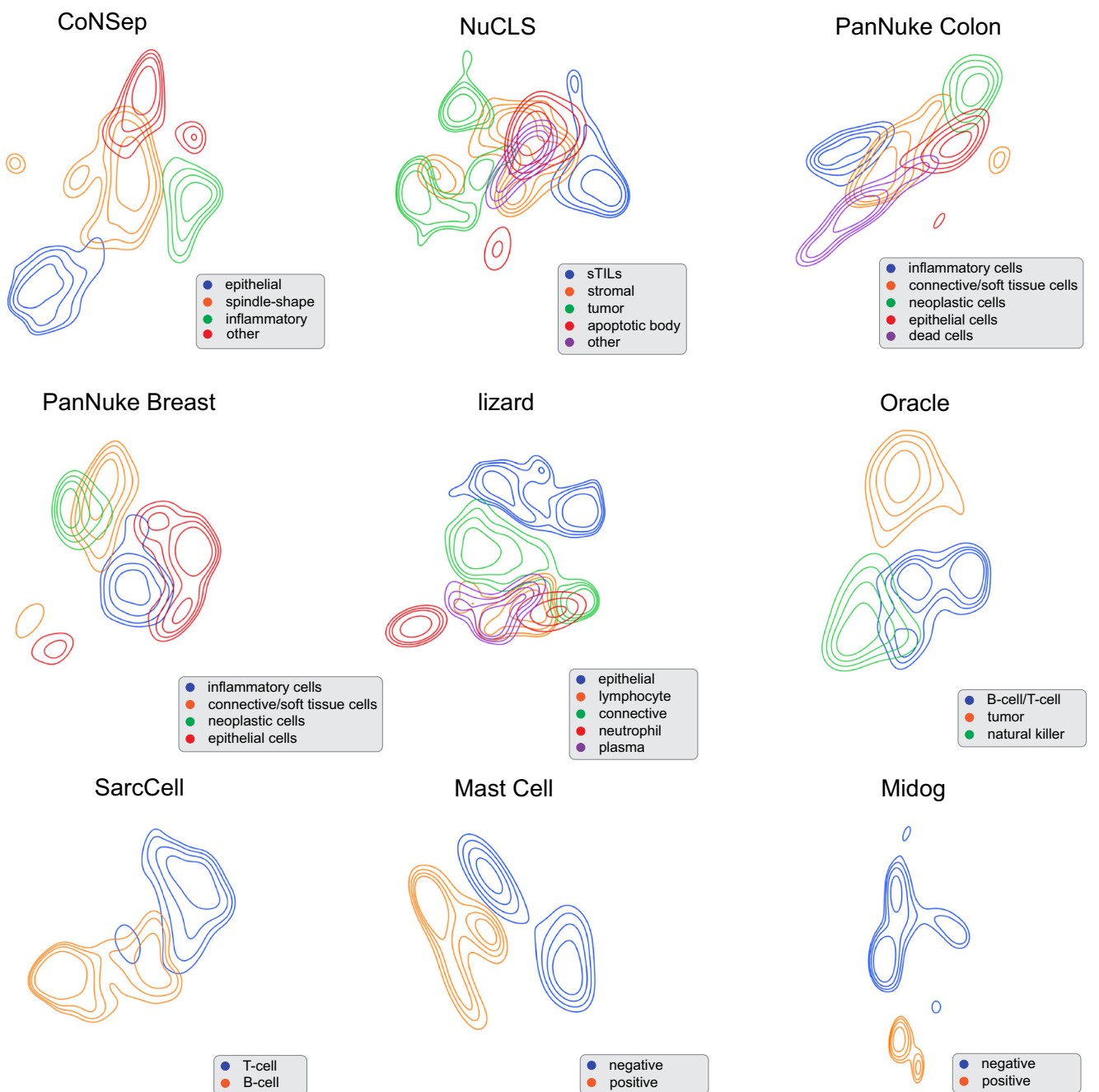

**Fig. 2 | Embedding space representation of each dataset using UMAP.** Embedding space representation of each dataset using UMAP. Contours with the same color demonstrate the distribution of the learned representations by our model for that specific cell types. Despite not using labeled data in the training process, our model learns to map cells with the same type close to each other. The co-centered contours with the same color show the distribution of the representation for cells with a specific type (Source data are provided as a Source Data file).

Additionally, we utilized a larger cohort of ovarian cancers containing 186 TMA cores to confirm our results in a larger scale. This cohort included two histotypes of epithelial ovarian cancers: high-grade serous and clear cell carcinomas. Following the same approach for patient clustering (as outlined above), we identified two major clusters (Fig. 4c) that were enriched with either the high-grade serous or clear cell carcinoma cases, suggesting a 92% accuracy in separating the two histotypes (14 of 186 that were mistakenly clustered in the wrong group).

To demonstrate the superiority of Volta for downstream analysis tasks compared to patch-based representation approaches, we employed a recent self-supervised model for patch representations[35].

Hierarchical clustering results assessed through the AMI, ARI, and Purity metrics (Supplementary Fig. 14 and Supplementary Table 14, Supplementary Note 10) demonstrate the superiority of clustering results of Volta compared to patched-based representation in downstream clustering of ovarian cancer histotypes.

We next demonstrated a potential application of VOLTA for exploratory cancer subtype discovery. More specifically, we scanned 19 whole-section slide images (WSI) corresponding to three molecular subtypes of endometrial cancer (EC): (1) DNA polymerase epsilon (POLE)-mutant cases, (2) cases with mismatch repair deficiency (MMRd), and (3) cases with p53 abnormality (p53abn) as assessed by immunohistochemistry. We next asked whether our proposed model

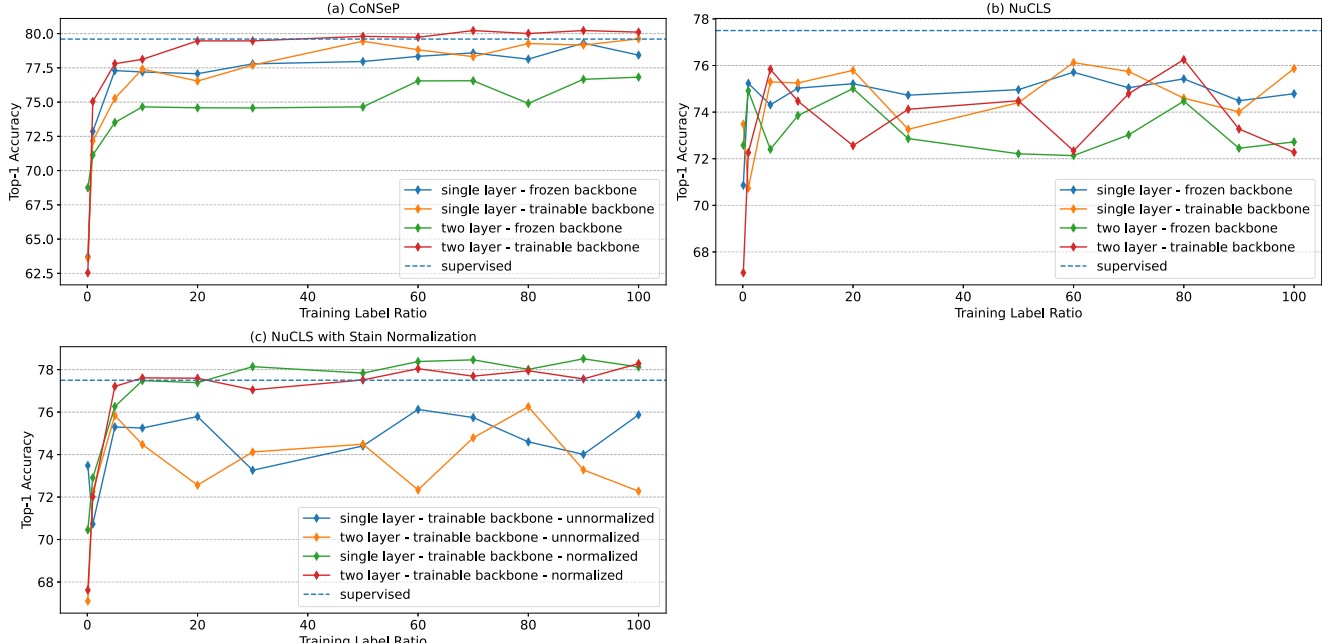

**Fig. 3 | Supervised fine-tuning results.** After pre-training using our self-supervised framework, a fully-connected layer (single- or double-layer) was added to the end of the backbone (the model generating the cell representations), and they were fine-tuned using the labeled data. We compared fine-tuning with both frozen and unfrozen backbone (**a** - CoNSeP and **b** - NuCLS). To account for the color differences in the train and test cohorts of the NuCLS dataset, we also performed the Vahedain color normalization before the fine-tuning process, which showed a significant boost compared to the unnormalized approach (**c**). The results demonstrate that our fine-tuned model can achieve the same performance as the supervised baselines (HoVer-Net and NuCLS) using only 20% of the labeled data while outperforming these baselines with the full set of the labeled data (**a** and **c**) (Source data are provided as a Source Data file).

could identify features in the H&E slides that would aid us in identifying the molecular subtypes of EC. After applying Volta and summarizing the features (Supplementary Note 10), we subjected EC WSI representations to clustering and identified three clusters of patients (Fig. 4b).

Interestingly, each of the three clusters was enriched with a specific molecular subtype of endometrial carcinoma. Similar to the procedure taken for the ovarian cancer dataset, we also visualized the cell clusters within the representative patches for each of the EC molecular subtypes (Supplementary Figs. 15–17) along with the cell cluster distributions (Supplementary Table 15 and Fig. 18). In line with recent findings, MMR-deficient tumors had the highest proportion of lymphocytes in the endometrial cancer dataset[50–52].

To further showcase the capability of the model on a larger scale dataset, we collected a cohort of patients with 633 TMA cores corresponding to the p53abn and NSMP (no specific molecular subtype) molecular subtypes of endometrial cancers. By taking the same approach as discussed above, we obtained two main clusters in the data (Fig. 4d) where each of the clusters was enriched with one of the two molecular subtypes. Furthermore, similar to the ovarian cancer dataset, we utilized the patch-based self-supervised learning baseline[35] to compare with Volta representations. Qualitative and quantitative results (Supplementary Fig. 14 and Supplementary Table 14) confirm the superiority of Volta compared to patch-based representation learning.

## Discussion

In this paper, we proposed a self-supervised framework (VOLTA) for learning cell representations from annotation-free H&E images. Our investigations confirm the superiority of VOLTA over the state-of-the-art models. Specifically, we demonstrated that VOLTA significantly outperformed the state-of-the-art unsupervised morphology- and deep-learning-based cell clustering methods on nine datasets, six cancer types, and datasets comprised of multiple cell types.

Utilizing unsupervised learning to generate cell representations introduces unique opportunities for discovery, prediction, and development purposes. For instance, as part of our experiments, we illustrated that VOLTA can be successfully used as a building block for cancer histotype clustering by applying it to two cohorts of ovarian (including 12 and 186 cases) as well as two cohorts of endometrical cancer (including 19 and 633 cases). Our findings are interesting from two aspects: 1) even though our model does not receive any patient labels at training time, it is able to identify clusters of patients that are similar to pathologist diagnosis or molecular subtypes; 2) VOLTA is data efficient to an extent that it worked on two datasets with 10–20 patients samples. This is in contrast to the commonly held notion that having a large dataset is usually a prerequisite for deep learning models. We also demonstrated that these improvements are not exclusive to the unsupervised aspects of the model but can also extend to a supervised setting. By using our pre-trained VOLTA as an initialization weight for a classification model, we achieved a performance equal to that of the state-of-the-art supervised models with as low as 10% of the labeled data, surpassing the state-of-the-art models with the full data. Additionally, we demonstrated that our self-supervised model is robust to undesired staining biases, which facilitates the utilization of a pre-trained model on datasets collected across different centers.

Our investigation has demonstrated the efficiency of VOLTA as a tool for cell discovery within multiple pathology pipelines. Leveraging a self-supervised framework, the model can be seamlessly integrated with a wealth of histopathology archives accessible from various clinical centers to enable the generation of extensive cell-level representation databases. Furthermore, the model has the potential to alleviate the laborious cell type labeling process by annotating cell clusters instead of individual cells and be used in an interactive pathology pipeline. In addition to its utilization in cell type discovery, we have also demonstrated that the model can serve as a foundational element for both histotype and molecular subtype identification. This

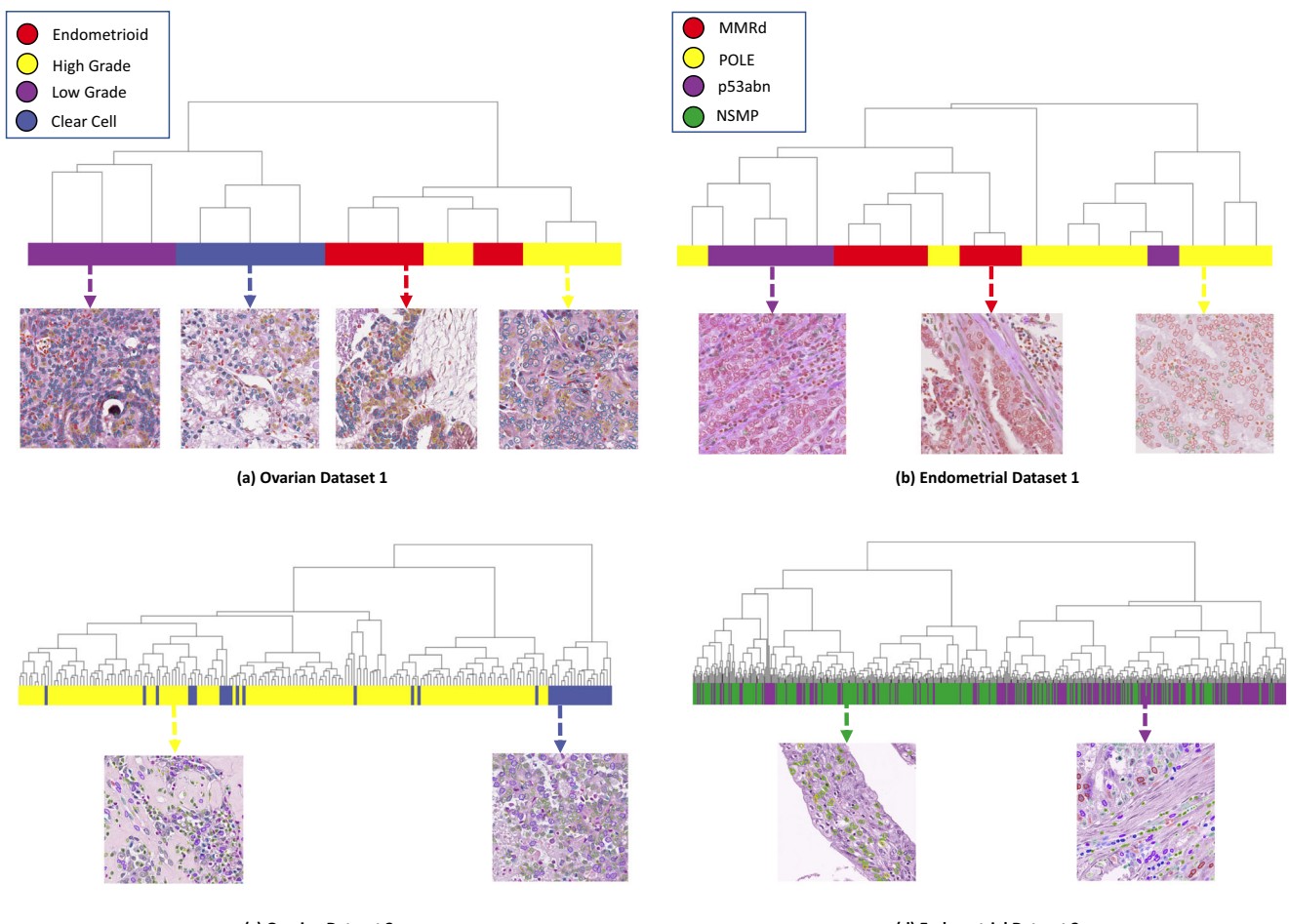

**Fig. 4 | Cancer subtype clustering across four datasets. a, c** Ovarian cancer and (**b, d**) endometrial cancer datasets are hierarchically clustered based on cell cluster proportions. To achieve this, we first train our model to deliver cell representations in a self-supervised manner. For the ovarian cases (**a, c**), our model will be applied to patches, a graph of cells will be built based on the cluster predictions, and the distribution of cell type clusters around each cell will be measured. Lastly, this distribution will be used to cluster the cases into distinct cohorts. In the case of endometrial cancer (**b, d**), we realize the cell count distribution across patches capture enough information for providing the separation. Therefore, after applying the model to each patch, we measure the distribution of cell type clusters across all the patches and use this distribution for a hierarchical clustering. In panel **b**, the supercluster on the right (yellow) demonstrates a cohort of patients that mostly have the POLE subtype (only one sample from p53abn is in this group), the supercluster in the middle (red) depicts mainly the MMRd patients (with only one POLE case misclassified), and the superclass on the left (purple) shows the p53abn cases with only one POLE case misplaced (Source data are provided as a Source Data file).

illustrates the wide-ranging potential of our model for discovery at multiple levels, from morphological features to molecular basis. These findings point to interesting directions for linking histopathology data to more advanced and in-depth areas such as genomic and molecular information.

The spatial distribution of cells within a tumor has been widely acknowledged to have a profound impact on the progression and prognosis of the disease. As demonstrated by[6], the bivariate analysis of immune and tumor cells can yield a wealth of information about the underlying biology of the disease. By utilizing metrics such as the Morisita-Horn index[53], Ripley's K function[54], and Intra-Tumor Lymphocyte Ratio (ITLR)[55], researchers have gained meaningful insights into the relationship between the spatial distribution of cells and clinical outcomes, identify immune-cancer hotspots, and predict chemotherapy response[32,56,57]. Considering the crucial role of cell identification in these applications, our research has the potential to be instrumental in enabling the aforementioned studies to be conducted at more extensive scales. This, in turn, can lead to a more profound understanding of the intricate correlation between disease phenotype and the spatial arrangement of the tumor microenvironment.

## Methods

### Ethics
The Declaration of Helsinki and the International Ethical Guidelines for Biomedical Research Involving Human Subjects were strictly adhered throughout the course of this study. All study protocols have been approved by the University of British Columbia/BC Cancer Research Ethics Board.

### Methodology
Fig. 1 provides an overview of the proposed self-supervised method for cell classification. This framework consists of two main blocks: 1) *Cell Block*; 2) *Environment Block*. The *Cell Block* learns the cell embeddings (i.e., representations) by contrasting individual cell-level images while the *Environment Block* incorporates environment-level information into the cell representations.

**Cell block.** The architectural design of the *Cell Block* is similar to our previously proposed model[58], which has shown promising performance in cell representation learning tasks. In this block, cell embeddings are learned by pulling the embeddings of two augmentations of the same image together, while the embeddings of other images are

pushed away. Let $\mathbf{X} = \{\mathbf{x}_i | 1 \le i \le N\}$ be the input batch of cell images and $N$ to be the number of images in the batch. Each $\mathbf{x}_i$ is a small crop of the H&E image around a cell in a way that it only includes that specific cell. Two different sets of augmentations are applied to $\mathbf{X}$ to generate $\mathbf{Q} = \{\mathbf{q}_i | 1 \le i \le N\}$ and $\mathbf{K} = \{\mathbf{k}_i | 1 \le i \le N\}$. We call these sets query and key, respectively. $\mathbf{q}_i$ and $\mathbf{k}_j$ are the augmentations of the same image if and only if $i = j$. The query batch is encoded using a backbone model, a neural network of choice, while the keys are encoded using a momentum encoder, which has the same architecture as the backbone. This momentum encoder is updated using (1) in which $\boldsymbol{\theta}_k^t$ is the parameter of momentum encoder at time $t$, $m$ is the momentum factor, and $\boldsymbol{\theta}_q^t$ is the parameter of the backbone at time $t$

$$\boldsymbol{\theta}_k^t = m\boldsymbol{\theta}_k^{t-1} + (1-m)\boldsymbol{\theta}_q^t. \tag{1}$$

Consequently, the obtained query and key representations are passed through separate Multi-Layer Perceptron (MLP) layers called projector heads. Although the query projector head is trainable, the key projector head is updated with momentum using the weight of the query projector head. We restrict these layers to be 2-layer MLPs with an input size of 512, a hidden size of 128, and an output size of 64. In addition to the projector head, we use an extra MLP on the query side of the framework, called the prediction head. This network is a 2-layer MLP with input, hidden, and output sizes of 64, 32, and 64, respectively. Similar to the last fully-connected layers of a conventional classification network, the projection and prediction heads provide more representation power to the model.

The networks of the *Cell Block* are trained using the InfoNCE[39] loss which is shown in (2)

$$\mathbf{L}_{q_i}^{cell} = -\log \frac{\exp \frac{\|f_q(\mathbf{q}_i)\|^2 \cdot \|f_k(\mathbf{k}_i)\|^2}{\tau}}{\sum_{j=0}^{N+Q} \exp \frac{\|f_q(\mathbf{q}_i)\|^2 \cdot \|f_k(\mathbf{k}_j)\|^2}{\tau}}. \tag{2}$$

In this equation, $\tau$ is the temperature that controls the sharpness of the distribution, $\|\,\|$ is the normalization operator, $Q$ is the number of items stored in the queue from the key branch, $f_q$ is the equal function for the combination of the backbone, query projection head, and query prediction head, and $f_k$ shows the equal function for the momentum encoder and the key projection head.

The augmentation pipelines include cropping, color jitter (brightness of 0.4, contrast of 0.4, saturation of 0.4, and hue of 0.1), gray-scale conversion, Gaussian blur (with a random sigma between 0.1 and 2.0), horizontal and vertical flip, and rotation (randomly selected between 0 to 180 degrees). To ensure the model consistently observes the entire cell image on one side, we eliminate the cropping step from one of the processes. Consequently, the pipeline that includes cropping generates localized sections of the cell image, while the other augmentation pipeline produces global images encompassing the complete view of the entire cell. Due to the randomness of augmentations, either one can be passed through the backbone or momentum-encoder.

Cell embeddings are generated from the trained momentum encoder at the inference time and are clustered by applying the K-means algorithm. One can use either the encoder or momentum encoder for embedding generation; however, the momentum encoder provides more robust representations since it aggregates the learned weights of the encoder network from all of the training steps (an ensembling version of the encoder throughout training)[33].

**Environment block.** Many studies have shown that the Tumor Micro Environment (TME) plays an important role in the tumor progression behavior[32,57]. Motivated by these findings, we ask: should the representation of a cell reflect its environment as well? Inspired by this question, we hypothesize that a deeper knowledge of the environment

leads to a better general understanding of the cell. In a mathematical formulation, this hypothesis is equivalent to the assumption that there exists mutual information between cells and their environment. Therefore, to validate this hypothesis, we propose to increase the mutual information between the corresponding cell and environment representations during the training process. Previous studies[59] have shown that the InfoNCE loss maximizes the lower bound of mutual information between different views of the image. Thus, we will use this loss function to achieve the aforementioned target by performing cross-modal contrastive learning as an auxiliary task.

Let $\mathbf{E} = \{\mathbf{e}_i | 1 \le i \le N\}$ be the corresponding environment patches of the cells represented by $\mathbf{X}$. Here, we refer to the environment as a large region around a cell in a way that includes the surrounding tissue and cells. Therefore, for $\forall\, i \in 1, 2, \ldots, N$, $\mathbf{x}_i$ and $\mathbf{e}_i$ are centered on the same cell (however, for the cases where the cells are located on the edge of the patch, we limit the patch border to the border of the image). After applying an augmentation pipeline, the environment patches are passed through an encoder network, called an environment encoder. Simultaneously, we apply a new projection head, the environment projection head, to the cell representations obtained from the query backbone in the *Cell Block*. Finally, one can train the *Environment Block* using these two sets of representations (environment and cell) and (3)

$$L_{q_i}^{env} = -\log \frac{\exp \frac{\|g_{cell}(\mathbf{q}_i)\|^2 \cdot \|g_{env}(\mathbf{e}_i)\|^2}{\tau}}{\sum_{j=0}^{N} \exp \frac{\|g_{cell}(\mathbf{q}_i)\|^2 \cdot \|g_{env}(\mathbf{e}_j)\|^2}{\tau}}. \tag{3}$$

Therefore, the final loss of the whole framework can be written as (4), in which $\lambda$ is a hyperparameter. Increasing the value of $\lambda$ prioritizes the mutual information of the cell with its environment over the consistency of the representation for different augmentations of the same cell

$$L_{q_i} = L_{q_i}^{cell} + \lambda L_{q_i}^{env}. \tag{4}$$

The augmentation pipeline of the *Environment Block* uses the same operations as that of the *Cell Block* except for cropping.

To prevent the model from focusing on the overlapping regions between the corresponding cell and environment images (called shortcut[60], meaning that the model uses undesired features to solve the problem), we mask the target cell in the environment patch. Furthermore, the rest of the cells in the environment patch are also masked to ensure that the model does not bias the representation of a cell towards the neighboring cell types. We will investigate the effectiveness of the masking operation in the ablation study.

### Data preparation

The aforementioned datasets included patch-level images, while we required cell-level ones for the training of the model. To generate such data, we used the instance segmentation provided in each of the external datasets to find cells and crop a small box around them. However, for the Oracle and SarcCell datasets, the instance segmentation masks were generated by applying HoVer-Net[21] segmentation pre-trained on the PanNuke dataset.

An adaptive window size was used to extract cell images from the H&E slides. More specifically, this window is selected based on the size of the cell, and this strategy is utilized to prevent overlapping with other cells. The adaptive window size was set to twice the size of the cell for the CoNSeP dataset while it was equal to the size of the cell for the rest of the datasets. Finally, cell images were resized to $32 \times 32$ pixels (to enable batch-wise processing operations) and were normalized to zero mean and unit standard deviation before being fed into our proposed framework. The environment patch used in the *Environment Block* was set to 200 pixels for all datasets.

Ground-truth label generation of the Oracle and SarcCell dataset cells was performed by finding the most expressed biomarker (by intensity and quantity) in the same position of the corresponding IHC image. To accommodate for the potential noise associated with image registration, two post-processing steps were performed: 1) the size of the window in the IHC image was set to 5 times of the window size in the H&E core (however, this scale was set to 1 for the SarcCell dataset due to more accurate co-registration performance); 2) the most expressed biomarker was considered as the label only if it contained at least 70% of the biomarker distribution in the IHC window.

## Implementation details

The code was implemented in Pytorch (v1.9.0), and the model was run on one and two V100 GPUs for the w/ and w/o environment settings, respectively. The batch size was set to 1024 (unless specified otherwise), the queue size to 65,536, and pre-activated ResNet18[61] was used for the backbone and momentum encoder in the *Cell Block*. The environment encoder architecture was set to LambdaNet model[62] as it extracts more informative patch representations using self-attention while keeping the computation and memory usage tractable. The stack was trained using the Adam optimizer for 500 epochs (unless specified otherwise) with a starting learning rate of 0.001, a cosine learning rate scheduler, and a weight decay of 0.0001. We also adopted a 10-epoch warm-up step. The momentum factor in the momentum encoders was 0.999, and the temperature was set to 0.07.

In Table 1 experiments, the training epoch count and batch size of our models were set to 200 and 512 for the PanNuke Breast, Lizard, Oracle, and SarcCell datasets. Additionally, for the training of our model on the Oracle datasets, we used 15,000 randomly selected cells from the training set, to reduce the training time.

In the self-supervised to supervised transfer learning step (cell classification), we adopted SGD (Stochastic Gradient Descent) with a starting learning rate of 0.001 using a cosine learning rate scheduler for 300 epochs with a batch size of 1024. Also, the weight decay was set to 0.00001. In the case that we allowed the encoder to be fine-tuned, we set the encoder's learning rate to 0.0001.

It is worth mentioning that for the cell classification of NuCLS, we followed the same super-class grouping of the original paper[22]. In this regard, we only used 3 super-classes out of 5 for cell type classification, including tumor, stromal, and sTILs.

## Baselines

The performance was also compared against five baselines. The pre-trained ImageNet model used weights that were pre-trained on the ImageNet dataset to generate the cell embeddings. The Morphological Features approach[63] adopted morphological features to produce a 30-dimensional feature vector, consisting of geometrical and shape attributes. Prior to clustering, the feature vectors were normalized to zero mean and unit standard deviation, and their size was reduced to 2 using t-SNE. The third baseline was Manual Feature[27] which used a combination of Scale-Invariant Feature Transform (SIFT) and Local Binary Patterns (LBP) features to provide representations for the cells. Similar to the previous baseline, we exercised standardization on the computed feature vectors. Additionally, our baseline set included two state-of-the-art unsupervised deep learning models. More specifically, the Auto-Encoder baseline adopted a deep convolution auto-encoder alongside a clustering layer to learn cell embeddings by performing an image reconstruction task[29]. And finally, the last baseline was GAN[27] which adopted the idea of InfoGAN[28] and developed a Generative Adversarial Network (GAN) for cell clustering by increasing the mutual information between the cell representation and a categorical noise vector.

## Statistics & reproducibility

The data selection and stratification were performed completely blind without any previous exposure to the patient or cell data. For public datasets, we used the train and test sets provided by the original publication; however, for the rest of the process, we took a completely blind approach.

The sample sizes used in this study are based on the sample provided sets from the original publication for the public datasets and the most available data for the private datasets. In both cases, we believe these sample sizes are sufficient for the study as at least 17,000 samples are available for each dataset.

Due to the stochastic nature of deep learning models, the exact reproduction of an experiment is not possible. However, we conducted each experiment multiple times and used the average of the results as the output.

## Reporting summary

Further information on research design is available in the Nature Portfolio Reporting Summary linked to this article.

## Data availability

The publicly available data used in this study (CoNSeP, NuCLS, Pan-Nuke, MiDOG, and Lizard datasets) are available in the original publications and their corresponding authors (https://arxiv.org/pdf/2204.03742, https://arxiv.org/pdf/1812.06499.pdf, https://arxiv.org/abs/2102.09099, https://arxiv.org/abs/2003.10778, https://arxiv.org/abs/2108.11195). The internal histopathology slides generated in this study (SarcCell, Oracle, and MastCell datasets) can be obtained by direct email to the corresponding author. All data accesses are subject to institutional permission and compliance with ethics from the corresponding institutions. Data can only be shared for non-commercial academic purposes and will require a data user agreement. The requested data will be provided as soon as all the corresponding institutions grant the required permissions. The rest of the data used for visualization purposes are included in the supplementary information. Source data are provided with this paper.

## Code availability

The code for this manuscript will be publicly available in https://github.com/AIMLab-UBC/VOLTA.

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

## Acknowledgements

This work was supported by Terry Fox Research Institute (J.N.M., A.B., grant number: 1116), Canadian Institute of Health Research (A.B., grant number: 201903PJT-418734), Natural Sciences and Engineering Research Council of Canada (A.B., grant number: RGPIN-2019-04896), Michael Smith Foundation for Health Research (A.B., grant number: SCH-2021-1546), Canada Research Chair (J.N.M., S.J.M.J.), Canada Foundation for Innovation/BC Knowledge Development Funds (AB, grant number: 41144), OVCARE Carraresi, and VGH UBC Hospital Foundation (A.B.). The funders had no involvement in study conception, data collection, data analysis, data interpretation, writing of the report, or publication decision.

## Author contributions

R.N. designed and benchmarked the models. A.D. initiated the study. R.N. and A.D. implemented the baseline models. R.N. and K.R. collected and pre-processed the data. R.N., A.B., and H.F. wrote the first draft of the manuscript. R.N., K.R., H.F., and A.B. revised the manuscript. A.Z. contributed to the pathology review of the model's results in terms of biological relevance. A.H. contributed to data analysis. J.N.M., S.J.M.J., C.B.G., B.H.N., S.T., K.M., E.S. contributed to cohort construction, tumor banking, experiments, pathology review, and computational infrastructure. A.B. and H.F. designed the experiments and supervised the study. A.B. conceived and oversaw the project and is the senior corresponding author. All authors have reviewed and approved the manuscript content.

## Competing interests

The authors declare no competing interests.
