## [Peer Review File · Nature Communications]

Reviewers' Comments:

Reviewer #1:

Remarks to the Author:

This manuscript presents a new technique to build representations for cells on H&E images. The method relies on self-supervision, which has not been widely used before in this context, and provides the benefit of not requiring large amounts of data or annotations. The method also learns from the environment of the cells. As a result, it achieves SOTA or near SOTA results in various cell classification tasks.

The work is an interesting technical development that will help computational pathology studies. However the scientific significance of the improvements are at this point still limited. Most of the tasks discussed in the manuscript, related to cell cluster definition, cell classification, and subtyping, already have high-quality SOTA results. An exploratory task focusing on classification of molecular subtypes of endometrial cancer is also presented, but it is not benchmarked against existing models, so it is not clear that VOLTA really offers an advantage in that case. New insights that could be provided by the model, for example related to interactions between the cells and their neighbourhoods, are not explored.

Detailed comments:

1. The datasets are not presented in detail. How many patients are there in each dataset, and how many slides? There may be significant biases between patients.
2. How does the model deal with artefacts? One of the biggest issues with unsupervised cell classification methods is that they capture artefacts, or cells that may have been affected by tissue processing, instead of real biological phenotypes. How clean were these datasets? Did you have instances of out of focus areas, bubbles, pen marks?
3. Section 2.1 presents results on the purity, AMI and ARI of different cell clusters. How did the numbers of clusters compare between the different methods?
4. Section 2.2 claims that VOLTA can distinguish between different immune cells. Is that unique to VOLTA?
5. One of the methodological benefits of VOLTA is supposed to be the fact that it captures the environment of the cell. What is the impact on tumour microenvironment measurements? E.g. does it identify variations in cellular phenotype when particular cells are close to each other? Or is it better able to identify spatial structure in the microenvironment, using any of the metrics discussed in the conclusion?

Reviewer #2:

Remarks to the Author:

The paper proposes a new topology for cell segmentation in H&E images based on self-supervision and contrastive learning. The most crucial idea put forward appears to be the simultaneous use of a cell image and an image of its immediate environment.

It is not clear where the acronym comes from. First encounter of an explanation says "Environment-Aware Contrastive Cell Representation Learning framework (VOLTA)" which is not clear and does not explain the acronym.

It is not motivated well why VOLTA needs two augmentations, and why one of them goes to the backbone. Why not three, or four? The augmentation operation is not specified anywhere.

Why should a contrastive learning between cell block and its environment proximity, using InfoNCE be a logical approach to the cell segmentation? Biologically, this is clear; topologically not motivated well.

The adaptive selection of cell blocks has not been further specified. How do you do it? What is the effect of resizing them to 32x32? What are the range of sizes?

How is the environment block selected? What size? Fixed size? Why that size? What is the effect of

a different size? Does the environment block size need to be set anew for a different primary sites?

An assessment of block size impact on the final results should be provided. This assessment should be based on experiments for different sizes, especially for environment block.

If Fig. 2 the best way to represent the results? (I assume the authors did that due to the large number of cells). Can a common "point" visualization be added as well?

Why is clustering evaluation the best approach to validate cell segmentation? Why AMI/ARI/Purity for cluster goodness? Why not silhouette coefficient or Dunn index?

Reviewer #3:

Remarks to the Author:

This paper describes a newly proposed self-supervised methods for learning visual representations using cell and its environment. The author showed good performance on multiple public datasets and a internal datasets.

The merit of the paper is the methods while the drawback is in experimental design and the layout. The network structure is novel and make use of the environmental information other than the cell image only.

However, the major issues are: 1) it did not compare to other self-supervised learning (SSL) methods for path-based classification problem which uses their internal dataset; 2) it did not compare other SSL methods for direct cell classification. Therefore, the purpose of this methods is lacking. The author did not use experiments to demonstrate why the two branch SSL structure proposed in this paper is necessary. Why don't we just use a off-the shelf SSL methods. One may also imagine that the robustness and generalization capability might be challenged by this complicated pipeline.

Also, as a medical image analysis paper for nature communications, the validation is lack of comprehensiveness. For example, the internal sample dataset, for the patch classification task, is too small to make any conclusion. Another example, in Fig 2., while U-map may show some separation for different types for cells, there is no comparisons to other methods that can be used extracting the feature representations. The qualitative analysis becomes pointless without comparison.

Last, the quality of writing does not meet nature standards, which lacks details on data generation, methods clarification. It reads fine for technical readers/domain readers. However, for most readers, it might be challenging to follow. Also, for many concepts/methods, citations are missing.

Below are the listed points the author should make changes on:

1. Line 092-093: 'Furthermore, to...has to be carried out again.'

I doubt this is always the case. For cell segmentation and phenotype, model may not need to be re-trained with newly labeled datasets.

2. Line 101-103: 'The former jeopardizes ... the accuracy of the model.'

I am not sure what the author was trying to say in terms of the context. Please rephrase/clarify.

3. Line 183-186: 'For instance, PanNuke Colon and CoNSeP.'

The author should provide that information for readers who are not in cell segmentation domain. Those acronyms can get readers totally lost. I understand that there is a section later to describe data, a brief description on the dataset and task is helpful to the reader especially those acronyms come first in the paper. Also, citation should be included.

4. Line 212-215: 'a reasonable performance ... achieved 80.2% and 76.3%.'

This statement is not that useful. The comparison does not make sense. The interpretation of reasonable performance is different from person to person. Especially the results are 62.7% and 72.6%. Those results are not reasonable to me at least. Instead, I feel you might say 'the performance achieves comparable performance when only 20% labeled data were used.'

5. In section 2.4.

Should the author perform heavier color augmentation can help on the staining variability?

6. Section 2.5.

The author spent a lot of efforts on this section while the results are constituted from a very small sample size with primarily qualitative results. I do not know if this kind of finding means anything. While we perform research discovery and this kind of finding may lead us to perform further experiments, it will not be adequate make any conclusion.

7. Line 328-334.

Refer to point 7.

8. Line 345-373.

A lot of random comments that is not related to the results and methods. The author can condense this into a few sentences for potential applications. Anyone can image any medical application.

9. Section 4.3.

The author did not describe the dataset clear enough. It is for in-domain readers. For other readers, they have no idea about the data. They do not even know what annotation, what task.

10. The author should cite and comment on this work:

'Identification of molecular cell type of breast cancer on digital histopathology images using deep learning and multiplexed fluorescence imaging.'

Reviewer 1 (R1)

R1 Comment 1: Most of the tasks discussed in the manuscript, related to cell cluster definition, cell classification, and subtyping, already have high-quality SOTA results.

Response: Thank you for this insightful comment. The main goal of this paper is to provide a framework for unsupervised cell representation learning without cell annotations or cell type labels. Therefore, we are mainly looking into unsupervised cell representations and cell clustering rather than (supervised) cell classification. This framework can then be utilized to perform downstream cell classification and subtyping. To the best of our knowledge, the stated baselines in Table 1 are the SOTA baselines for cell clustering, and Volta is outperforming all these baselines in most of the datasets and tissue types, having slightly inferior performance only on Oracle and SarcCell datasets. Furthermore, we have also added two new baselines to our comparisons: SimCLR [Chen et al. 2020] and DINO [Caron et al. 2021]. In particular, Volta outperforms all of seven baselines across nine datasets and three metrics by a significant margin (CoNSEP, NuCLS, PanNuke Breast, PanNuke Colon, Lizard, MastCell, and MiDOG), while it works on par with SimCLR and GAN on the Oracle and SarcCell datasets, respectively (Table 1). Moreover, by using Volta as a pre-training stage for supervised cell classification, we showed that the model can outperform the SOTA supervised baselines with the full labeled dataset, while it achieves the same performance with only 20% of the labeled data (Figure 3). Additionally, for the task of unsupervised subtype clustering, we added a new baseline and showed the superior performance of Volta compared to it. Please see our response to R1 Comment 2 for further details.

Table 1 Unsupervised clustering of cell representations across different methods and datasets. The baseline models include both morphology-based and state-of-the-art deep learning methods for cell representation. Some of the baseline results are listed as ”-” meaning calculation of the feature vectors was not possible due to the limitation of the model on the small-sized cells.

Model	Metric	CoNSEP	NuCLS	PanNuke Breast	PanNuke Colon	Lizard	Oracle	SarcCell	MastCell	MiDOG
Pre-trained ImageNet	AMI	7.3%	9.3%	5.42%	11.21%	6.25%	0.26%	0.8%	0.1 %	13.1%
	ARI	7%	7.8%	3.94%	8.21%	4.36%	0.42%	1.8%	0.1 %	5.8 %
	Purity	42.7%	56.7%	41.15%	43.93%	50.4%	48.87%	42.0%	58.1%	62.1%
Morphological	AMI	12.7%	21.1%	8.94%	7.88%	13.21%	-	-	0.0 %	-
	ARI	1.3%	18.8%	7.28%	6.19%	9.22%	-	-	0.0 %	-
	Purity	48.8%	66.1%	47.06%	42.73%	57.5%	-	-	58.1%	-
Manual Features	AMI	9.5%	11.25%	-	7.86%	10.2%	2.74%	2.9%	2.1 %	6.1%
	ARI	6.4%	7.8%	-	6.53%	3.8%	2.24%	2.1%	4.3 %	7.4%
	Purity	45.5%	56.2%	-	40.37%	52.9%	53.84%	42.7%	62.0%	63.7%
DCAE	AMI	10.1%	8.3%	6.41%	11.43%	4.36%	3.93%	-%	0.0 %	3.5%
	ARI	7.3%	7.2%	5.11%	10.01%	2.34%	3.84%	-%	0.0 %	4.3%
	Purity	50.5%	56.8%	43.49%	45.18%	49.38%	58.69%	-%	58.1%	60.5%
GAN	AMI	14.8%	14%	6.7%	13.7%	7.5%	4.1%	6.0%	0 %	21.4%
	ARI	15.7%	12.6%	4.6%	11.4%	3%	5.8%	5.6%	0 %	27.9%
	Purity	58.4%	62%	42.4%	49.6%	48.9%	57.5%	46.0%	58.0%	76.5%
SimCLR	AMI	19.6%	20.1%	10.7%	13.9%	16.5%	12.5%	5.6%	6.2 %	30.2%
	ARI	16.7%	22.1%	8.6%	8.9%	11.1%	14.2%	4.5%	8.4 %	30.7%
	Purity	57.5%	68.2%	48.3%	40.9%	57.1%	67.5%	45.2%	65.1%	77.7%
DINO	AMI	1.9%	0.6%	0.3%	7.5%	0.4%	0.4%	0.3%	0.0 %	4.9%
	ARI	1.7%	0.7%	0.6%	5.5%	0.0%	0.5%	0.8%	0.5 %	6.6%
	Purity	1.9%	0.7%	0.5%	7.1%	0.4%	0.6%	41.9%	58.1%	62.9%
Volta (w/o env)	AMI	24.2%	22.8%	10.75%	19.5%	10.85%	3.6%	5.3%	0 %	35.5%
	ARI	21.7%	24%	7.58%	16.1%	6.2%	2.45%	3.7%	0 %	39.4%
	Purity	51.3%	68.3%	46.87%	54.6%	52.66%	54.7%	43.8%	58.1%	81.4%
VOLTA	AMI	25.5%	26.2%	13.8%	22.5%	17.3%	8.05%	4.2%	25.4%	50.4%
	ARI	19.3%	27.3%	8.94%	21.8%	11.4%	4.95%	6.7%	33.1%	60.3%
	Purity	63.5%	70.3%	47.7%	56.9%	57.9%	59.45%	44.8%	79.0%	88.8%

Fig. 3 After pre-training using our self-supervised framework, a fully-connected layer (single- or double-layer) was added to the end of the backbone (the model generating the cell representations), and they were fine-tuned using the labeled data. We compared fine-tuning with both frozen and unfrozen backbone (a - CoNSeP and b - NuCLS). To account for the color differences in the train and test cohorts of the NuCLS dataset, we also performed the Vahedain color normalization before the fine-tuning process, which showed a significant boost compared to the unnormalized approach (c). The results demonstrate that our fine-tuned model can achieve the same performance as the supervised baselines (HoVer-Net and NuCLS) using only 20% of the labeled data while outperforming these baselines with the full set of the labeled data (a and c).

R1 Comment 2: An exploratory task focusing on classification of molecular subtypes of endometrial cancer is also presented, but it is not benchmarked against existing models, so it is not clear that VOLTA really offers an advantage in that case.

Response: This is an excellent suggestion for which we thank the reviewer. We would like to note that our focus has been on unsupervised cell clustering and their utility to perform downstream tasks with clinical or biological utility. That said, we also agree with the reviewer that benchmarking against existing methods in downstream tasks would also be beneficial. Although supervised molecular subtype classification of endometrial cancer has been investigated previously [Fremont et al. 2023], to the best of our knowledge, the task of unsupervised clustering of endometrial cancer and their correlation with molecular subtypes has not been investigated before. Nevertheless, to address this concern, we utilized a SOTA patch-based self-supervised model [Ciga et al. 2022] to perform a similar task (i.e., clustering endometrial cancer cases). Furthermore, we added two large datasets of ovarian and endometrial cancer with 186 (95 unique patients) and 633 (633 unique patients) cases, respectively. Supplementary Table A13 and Supplementary Figure A5 show the quantitative and qualitative improvements of Volta compared to this baseline on these large datasets. We would like to highlight that, our proposed approach (i.e., unsupervised cell representation and simple aggregations of cell representations) requires significantly less data points compared to supervised classification tasks where we require a large dataset to train deep learning models. For instance, in this work, we have demonstrated that Volta can enable the molecular subtype clustering with only a few data samples (~19 cases), while a supervised approach requires much more data points.

Table A13 Comparison of subtype classification for Volta and Patch-based SimCLR.

Dataset	AMI	ARI	Purity
Ovarian Datasets			
VOLTA (ours)	61.96%	72.09%	94.09%
Patch-Based SimCLR	22.53	14.68	80.65
Endometrial Datasets			
VOLTA (ours)	4.50%	5.54%	61.93%
Patch-Based SimCLR	0.99	1.66	56.76

Fig. A5 Qualitative comparison of cancer subtype clustering between VOLTA and patch-based SSL Ciga et al (2022) on ovarian dataset 2 and endometrial dataset 2

R1 Comment 3: New insights that could be provided by the model, for example related to interactions between the cells and their neighborhoods, are not explored.

Response: Thank you for the comment. To assess this, we have removed the environment block of the model and then compared the results with the proposed version of Volta with environment block (Table 1). The results demonstrate that the presence of the environment block can positively impact the clustering performance of the model, highlighting the mutual interactions between the cells and their environment. Additionally, our experiments show that increasing the size of the environment can increase the clustering performance, while too large environment patches can also reduce this performance to some extent (Supplementary Table A11).

Table 1 Unsupervised clustering of cell representations across different methods and datasets. The baseline models include both morphology-based and state-of-the-art deep learning methods for cell representation. Some of the baseline results are listed as ”-” meaning calculation of the feature vectors was not possible due to the limitation of the model on the small-sized cells.

Model	Metric	CoNSEP	NuCLS	PanNuke Breast	PanNuke Colon	Lizard	Oracle	SarcCell	MastCell	MiDOG
Pre-trained ImageNet	AMI	7.3%	9.3%	5.42%	11.21%	6.25%	0.26%	0.8%	0.1 %	13.1%
	ARI	7%	7.8%	3.94%	8.21%	4.36%	0.42%	1.8%	0.1 %	5.8 %
	Purity	42.7%	56.7%	41.15%	43.93%	50.4%	48.87%	42.0%	58.1%	62.1%
Morphological	AMI	12.7%	21.1%	8.94%	7.88%	13.21%	-	-	0.0 %	-
	ARI	1.3%	18.8%	7.28%	6.19%	9.22%	-	-	0.0 %	-
	Purity	48.8%	66.1%	47.06%	42.73%	57.5%	-	-	58.1%	-
Manual Features	AMI	9.5%	11.25%	-	7.86%	10.2%	2.74%	2.9%	2.1 %	6.1%
	ARI	6.4%	7.8%	-	6.53%	3.8%	2.24%	2.1%	4.3 %	7.4%
	Purity	45.5%	56.2%	-	40.37%	52.9%	53.84%	42.7%	62.0%	63.7%
DCAE	AMI	10.1%	8.3%	6.41%	11.43%	4.36%	3.93%	-%	0.0 %	3.5%
	ARI	7.3%	7.2%	5.11%	10.01%	2.34%	3.84%	-%	0.0 %	4.3%
	Purity	50.5%	56.8%	43.49%	45.18%	49.38%	58.69%	-%	58.1%	60.5%
GAN	AMI	14.8%	14%	6.7%	13.7%	7.5%	4.1%	6.0%	0 %	21.4%
	ARI	15.7%	12.6%	4.6%	11.4%	3%	5.8%	5.6%	0 %	27.9%
	Purity	58.4%	62%	42.4%	49.6%	48.9%	57.5%	46.0%	58.0%	76.5%
SimCLR	AMI	19.6%	20.1%	10.7%	13.9%	16.5%	12.5%	5.6%	6.2 %	30.2%
	ARI	16.7%	22.1%	8.6%	8.9%	11.1%	14.2%	4.5%	8.4 %	30.7%
	Purity	57.5%	68.2%	48.3%	40.9%	57.1%	67.5%	45.2%	65.1%	77.7%
DINO	AMI	1.9%	0.6%	0.3%	7.5%	0.4%	0.4%	0.3%	0.0 %	4.9%
	ARI	1.7%	0.7%	0.6%	5.5%	0.0%	0.5%	0.8%	0.5 %	6.6%
	Purity	1.9%	0.7%	0.5%	7.1%	0.4%	0.6%	41.9%	58.1%	62.9%
Volta (w/o env)	AMI	24.2%	22.8%	10.75%	19.5%	10.85%	3.6%	5.3%	0 %	35.5%
	ARI	21.7%	24%	7.58%	16.1%	6.2%	2.45%	3.7%	0 %	39.4%
	Purity	51.3%	68.3%	46.87%	54.6%	52.66%	54.7%	43.8%	58.1%	81.4%
VOLTA	AMI	25.5%	26.2%	13.8%	22.5%	17.3%	8.05%	4.2%	25.4%	50.4%
	ARI	19.3%	27.3%	8.94%	21.8%	11.4%	4.95%	6.7%	33.1%	60.3%
	Purity	63.5%	70.3%	47.7%	56.9%	57.9%	59.45%	44.8%	79.0%	88.8%

Table A11 Effect of environment patch size on the cell clustering performance of the model evaluated on the CoNSEP dataset. Env 50, Env 100, Env 200, and Env 300 represent the experiments conducted by the environment patch size of 50, 100, 200, and 300 pixels, respectively.

Model	Metric	ENV 50	ENV 100	ENV 200	ENV 300
VOLTA	AMI	14.3%	17.1%	25.5%	23.7%
	ARI	6.0%	10.7%	19.3%	14.7%
	Purity	50.1%	51.7%	63.5%	60.0%

R1 Comment 4: The datasets are not presented in detail. How many patients are there in each dataset, and how many slides? There may be significant biases between patients.

Response: Done. As requested, this information has been added to Section A.1 of the supplementary text (page 26 and lines 1162-1288). Please note this information is not available for some of the public datasets. As an example, the Pannuke and Lizard datasets have been gathered from multiple centers and the original manuscripts state that they included a large number of patients. However, the exact number of patients or slides has not been reported for them.

R1 Comment 5: How does the model deal with artefacts? One of the biggest issues with unsupervised cell classification methods is that they capture artefacts, or cells that may have been affected by tissue processing, instead of real biological phenotypes. How clean were these datasets? Did you have instances of out of focus areas, bubbles, pen marks?

Response: To our knowledge, there are not significant artifacts such as wrinkles, crushed tissue, bubbles, or out of focus areas in our datasets. Specifically, the publicly available datasets have been annotated by pathologists and have been extensively utilized for benchmarking experiments in the field [Doan et al. 2022, Jaume et al. 2021, Ryu et al. 2023]. The private datasets were also visually inspected to minimize and exclude such artifacts (we also added this to the description of each datasets in the Supplementary Section A.1 in page 26 and lines 1162-1288). Additionally, the cell clusters generated by Volta have been visually inspected by a pathologist to confirm the biological validity of the identified cell clusters.

R1 Comment 6: Section 2.1 presents results on the purity, AMI and ARI of different cell clusters. How did the numbers of clusters compare between the different methods?

Response: Thank you for pointing this out. Given that in section 2.1, the ground truth cell labels are known (as well as the total number of ground truth cell clusters), we used the same number of clusters to divide the model predictions. More specifically, the cell representations are learned through an unsupervised approach (not using the ground truth labels or their count), and then they are divided into N clusters where N is the number of ground truth cell clusters (i.e., number of cell types). This number for each dataset is shown in Supplementary Table A1. We also added this to page 30 and lines 1349-1352 of the supplementary for more clarification.

Table A1 Datasets summary. The datasets are distributed across 4 tissue and 18 cell types to demonstrate the utility of the method.

Dataset	Cell Type Count	Tissue Type	Magnification Scale	Total Number of Cells	Cell Types	Number of Institutions
CoNSeP	4	Colon	40x	24,319	Spindle, epithelial, inflammatory, miscellaneous	1
NuCLS	5	Breast	20x	51,986	Tumor, stromal, sTILs, apoptotic, miscellaneous	18
PanNuke Breast	4	Breast	40x	34,806	Neoplastic, epithelial, inflammatory, and connective	> 2
PanNuke Colon	5	Colon	40x	23,831	Neoplastic, epithelial, inflammatory, connective, and dead	> 2
Lizard	6	Colon	20x	277,654	Epithelial, connective tissue, lymphocyte, plasma, neutrophil, and eosinophil	> 7
Oracle	3	Ovarian	40x	265,580	Tumor, T-cell, B-cell, and Natural Killer	1
SarcCell	2	Soft Tissue	40x	65,403	T-cell, B-cell, and dendritic	1
Mast Cell	2	Skin	40x	20,334	Mast Cell, miscellaneous	1
MiDOG	2	Dog Lung, Dog Lymphoma, Human Breast, Human Neuroendocrine	40x	17,081	Mitotic, miscellaneous	1

R1 Comment 7: Section 2.2 claims that VOLTA can distinguish between different immune cells. Is that unique to VOLTA?

Response: Thank you for the insightful comment. We have included the UMAP plots of some of the best performing baselines in Section A.2.4 of the supplemental. Our results demonstrate that SimCLR can also separate the B-Cell and T-Cell from each other within the context of sarcoma cases (SarcCell). While it's challenging to determine which model performs better qualitatively based on Figure 2 and Supplementary Figure A4 (included below), our quantitative results (found in Table 1 on page 3 of this document) demonstrate that they perform equally well.

Fig. 2 Embedding space representation of each dataset using UMAP. Contours with the same color demonstrate the distribution of the learned representations by our model for that specific cell types. Despite not using labeled data in the training process, our model learns to map cells with the same type close to each other. The co-centered contours with the same color show the distribution of the representation for cells with a specific type.

Fig. A4 The UMAP visualization of SimCLR baseline model for different dataset

R1 Comment 8: One of the methodological benefits of VOLTA is supposed to be the fact that it captures the environment of the cell. What is the impact on tumour microenvironment measurements? E.g. does it identify variations in cellular phenotype when particular cells are close to each other? Or is it better able to identify spatial structure in the microenvironment, using any of the metrics discussed in the conclusion?

Response: Thank you for highlighting this. One of the benefits of Volta is incorporating the environment information in cell representation learning. To show this, we remove the environment block of Volta and measure the same metrics for the trained model. These experiments are performed for all of the datasets, and the results highlighted in the “Volta (w/o env)” row of Table 1 (included in page 3 of this document) demonstrate that the incorporation of the environment can improve the clustering metrics consistently. We believe this improvement is mainly due to the spatial structure of the environment as all the cells within the environment patch are masked.

Reviewer (R2)

R2 Comment 1: It is not clear where the acronym comes from. First encounter of an explanation says “Environment-Aware Contrastive Cell Representation Learning framework (VOLTA)” which is not clear and does not explain the acronym.

Response: Done. The VOLTA acronym comes from the “enVironment aware cOntrastive cell representatiOn leArning”. We have now highlighted this in the abstract (page 1. Line 40) and the manuscript (page 4, line 157-158) as well as the title

R2 Comment 2: It is not motivated well why VOLTA needs two augmentations, and why one of them goes to the backbone. Why not three, or four? The augmentation operation is not specified anywhere.

Response: Thank you for the comment. This structure is inspired from the architectural design of self-supervised models [Chen et al. 2020]. The main purpose of doing so is to have two visually different looking images of the exact same cells. Even though it is possible to utilize more than two branches (i.e., more than two sets of augmentations), using two branches is the simplest design and prevents the complications in the pipeline and especially the loss function. Since these augmentations are random, either one can be passed through the backbone. We have also revised the manuscript in section 2.1, page 4, lines 161-164 and lines 168-170, to elaborate more on this aspect. However, more details can be found in [LeCun et al. 2006].

Additionally, we have highlighted the details of the augmentations in the methods section page 11, at lines 485-494 and rephrased them for more clarity and intuition behind having two branches. Specifically, the two augmentation pipelines include cropping, color jitter (brightness of 0.4, contrast of 0.4, saturation of 0.4, and hue of 0.1), gray-scale conversion, Gaussian blur (with a random sigma between 0.1 and 2.0), horizontal and vertical flip, and rotation (randomly selected between 0 to 180 degrees) operations. To ensure the model always views the whole-cell image of the cell on one side, we remove the cropping operation from one of the pipelines. Therefore, the pipeline with cropping generates local regions of the cell image while the images generated by the other augmentation pipeline are global, containing the whole-cell view.

R2 Comment 3: Why should a contrastive learning between cell block and its environment proximity, using InfoNCE be a logical approach to the cell segmentation? Biologically, this is clear; topologically not motivated well.

Response: The topological design stems from our initial hypothesis. Specifically, we hypothesize that there is mutual information between a cell (i.e., a small crop around the cell) and its environment (i.e., a large patch around the cell including the environment and other cells). Therefore, during training, we would like to increase this mutual information. To do so, we use the InfoNCE loss, which has been shown to increase the lower bound of mutual information (thus the mutual information itself) between the query and the positive sample. This is the main motivation for the design. We have also revised the manuscript on page 4 and lines 173-175 to reflect this.

R2 Comment 4: The adaptive selection of cell blocks has not been further specified. How do you do it? What is the effect of resizing them to 32x32? What are the range of sizes?

Response: Thank you for the comment. Adaptive window size means that the window is selected based on the size of the cell. This strategy has been taken to make sure that the window contains an isolated cell. The resizing operation is necessary as it will enable computation parallelization to process images in batch. Given that this transformation is applied to all the cells in the same way, cells with a specific morphology would still end up with similar morphologies, after the transformation. The original images have a size of 30x30 pixels, prior to the rescaling operation. Our extensive experimentations and visual inspection of the identified cell clusters by pathologists show that Volta effectively identified various cell types. We have modified page 13, lines 561-568 of the manuscript to clarify this.

R2 Comment 5: How is the environment block selected? What size? Fixed size? Why that size? What is the effect of a different size? Does the environment block size need to be set anew for a different primary sites? An assessment of block size impact on the final results should be provided. This assessment should be based on experiments for different sizes, especially for environment block.

Response: Thank you for this insightful comment. The environment size is set to 200 pixels for all datasets. This number is fixed for all datasets and data samples. However, to expand more on this, we performed experiments with various sizes of environment patches (Supplementary Table A11). Our results demonstrate that the performance of the model improves by increasing the environment patch size. However, too large environment patches can also result in a suboptimal performance.

Table A11 Effect of environment patch size on the cell clustering performance of the model evaluated on the CoNSeP dataset. Env 50, Env 100, Env 200, and Env 300 represent the experiments conducted by the environment patch size of 50, 100, 200, and 300 pixels, respectively.

Model	Metric	ENV 50	ENV 100	ENV 200	ENV 300
VOLTA	AMI	14.3%	17.1%	25.5%	23.7%
	ARI	6.0%	10.7%	19.3%	14.7%
	Purity	50.1%	51.7%	63.5%	60.0%

R2 Comment 6: Is Fig. 2 the best way to represent the results? (I assume the authors did that due to the large number of cells). Can a common “point” visualization be added as well?

Response: Thank you for the comment. Given the large number of cells within each dataset, the point visualization of UMAPs have many overlapping cells, making the visual interpretation difficult. Therefore, we used the contour plots to show the densities of cell populations within each subtype. However, to address this comment, we have also added the point visualizations in Figure A1.

Fig. A1 The point-wise UMAP visualization of VOLTA for different dataset

R2 Comment 7: Why is clustering evaluation the best approach to validate cell segmentation? Why AMI/ARI/Purity for cluster goodness? Why not silhouette coefficient or Dunn index?

Response: Thank you for this insightful comment. Just to clarify, the main purpose of this work is to provide a framework for unsupervised cell representation learning. The best way to demonstrate the performance of the model is to see if the representations from the same cell type are closer to each other or not. Therefore, cell clustering is used as the main approach for doing so, similar to previous publications in this context [He et al. 2018].

To the best of our knowledge, Purity/AMI/ARI are the most commonly used metrics for evaluation of clustering methods, and they can evaluate the model from different aspects instead of just one. Even though silhouette and Dunn index metrics are also used, they are more focused on assessing the internal cohesion and separation of clusters in a point-wise manner. More specifically, Dunn index calculates the minimum distance between the points from two clusters and the maximum distances between the points within each cluster. Given that some of the cells are inevitably categorized within a different cluster (even the misclassification of a single cell can cause this issue), the minimum distance will be equal to 0, which results in the Dunn Index to be equal to 0. Nevertheless, we have also calculated Silhouette coefficient and Dunn index (Table A2).

Table A2 Unsupervised clustering of cell representations across different methods and datasets in terms of Dunn Index and Silhouette Score.

Model	Metric	CoNSeP	NuCLS	PanNuke Breast	PanNuke Colon	Lizard	Oracle	SarcCell	MastCell	MiDOG
Pre-trained ImageNet	Dunn Index	0.0%	0.0%	0.0%	0.0%	-	0.0%	0.0%	0.0%	0.0%
	Silhouette	-1.2%	-11.6%	-5.3%	-2.6%	-	-0.2%	2.9%	0.0%	19.0%
Morphological	Dunn Index	0.0%	0.0%	0.0%	0.0%	0.0%	0.0%	0.0%	0.0%	0.0%
	Silhouette	-1.6%	-6.8%	-11.9%	-7.0%	-7.6%	0.7%	4.8%	-1.0%	0.0%
Manual Features	Dunn Index	0.0%	0.0%	0.0%	0.0%	0.0%	0.0%	0.0%	0.0%	0.0%
	Silhouette	-15.0%	-6.6%	-7.5%	-4.0%	-10.0%	-6.7%	1.2%	0.0%	13.1%
DCAE	Dunn Index	0.0%	0.0%	0.0%	0.0%	-	-	-	0.0%	0.0%
	Silhouette	0.8%	-5.7%	-1.4%	-1.0%	-	-	-	3.8%	5.0%
SimCLR	Dunn Index	0.0%	0.0%	0.0%	0.0%	-	0.0%	0.0%	0.0%	0.0%
	Silhouette	2.5%	-2.1%	-0.6%	-1.9%	-	2.2%	5.1%	1.4	14.2
DINO	Dunn Index	0.0%	0.0%	0.0%	0.0%	-	-	0.0%	0.0%	0.0%
	Silhouette	-2.0%	-10.0%	-3.4%	-4.0%	-	-	2.5%	2.7%	12.1%
Volta (w/o Env)	Dunn Index	0.0%	0.0%	0.0%	0.0%	-	0.0%	0.0%	0.0	0.0
	Silhouette	2.6%	-3.3%	-5.0%	-6.3%	-	0.2%	3.7%	21.2%	23.0%
VOLTA	Dunn Index	0.0%	0.0%	0.0%	0.0%	0.0%	0.0%	0.0%	0.0%	0.0%
	Silhouette	4.0%	-3.3%	-3.2%	-1.8%	2.1%	3.3%	2.7%	11.2%	12.1%

Reviewer 3 (R3)

R3, Comment 1: it did not compare to other self-supervised learning (SSL) methods for patch-based classification problem which uses their internal dataset

Response: Thank you for this valuable comment. As per reviewer's suggestion, we compared our cell-based representation model with self-supervised models for patch-based classification. More specifically, we utilized a pretrained SSL ResNet18 model from [Ciga et al 2022] and applied it to the ovarian histotype and endometrial molecular subtype clustering. Specifically, we extracted features from the model for 224×224 pixel patches and averaged these representations for the patches from the same core, and finally applied a hierarchical algorithm on top of these representations. The results can be found in Supplementary Figure A5 and Table A.13 (below). Our findings suggest that the learned representations of Volta can result in a better histotype and molecular subtype clustering compared to the patch-based SSL method. These findings are supported by both the aforementioned qualitative and quantitative results.

Fig. A5 Qualitative comparison of cancer subtype clustering between VOLTA and patch-based SSL Ciga et al (2022) on ovarian dataset 2 and endometrial dataset 2

Table A13 Comparison of subtype classification for Volta and Patch-based SimCLR.

Dataset	AMI	ARI	Purity
Ovarian Datasets			
VOLTA (ours)	61.96%	72.09%	94.09%
Patch-Based SimCLR	22.53	14.68	80.65
Endometrial Datasets			
VOLTA (ours)	4.50%	5.54%	61.93%
Patch-Based SimCLR	0.99	1.66	56.76

R3, Comment 2: it did not compare other SSL methods for direct cell classification. Therefore, the purpose of this methods is lacking.

Response: Thank you for this constructive comment. As suggested by the reviewer, we have now included comparisons with other SSL methods, namely SimCLR and DINO for cell representation. We have summarized the results in Table 1 (included below). Our results demonstrate that Volta can still outperform the added baselines on majority of the datasets. More specifically, Volta has outperformed SimCLR and Dino on NuCLS, CoNSEP, Pannuke Breast, Pannuke Colon, Lizard, MastCell, and MiDOG dataset, while SimCLR performs slightly better on the Oracle dataset only along with the Purity metric for the Pannuke Breast dataset.

Table 1 Unsupervised clustering of cell representations across different methods and datasets. The baseline models include both morphology-based and state-of-the-art deep learning methods for cell representation. Some of the baseline results are listed as ”-” meaning calculation of the feature vectors was not possible due to the limitation of the model on the small-sized cells.

Model	Metric	CoNSEP	NuCLS	PanNuke Breast	PanNuke Colon	Lizard	Oracle	SarcCell	MastCell	MiDOG
Pre-trained ImageNet	AMI	7.3%	9.3%	5.42%	11.21%	6.25%	0.26%	0.8%	0.1 %	13.1%
	ARI	7%	7.8%	3.94%	8.21%	4.36%	0.42%	1.8%	0.1 %	5.8 %
	Purity	42.7%	56.7%	41.15%	43.93%	50.4%	48.87%	42.0%	58.1%	62.1%
Morphological	AMI	12.7%	21.1%	8.94%	7.88%	13.21%	-	-	0.0 %	-
	ARI	1.3%	18.8%	7.28%	6.19%	9.22%	-	-	0.0 %	-
	Purity	48.8%	66.1%	47.06%	42.73%	57.5%	-	-	58.1%	-
Manual Features	AMI	9.5%	11.25%	-	7.86%	10.2%	2.74%	2.9%	2.1 %	6.1%
	ARI	6.4%	7.8%	-	6.53%	3.8%	2.24%	2.1%	4.3 %	7.4%
	Purity	45.5%	56.2%	-	40.37%	52.9%	53.84%	42.7%	62.0%	63.7%
DCAE	AMI	10.1%	8.3%	6.41%	11.43%	4.36%	3.93%	-	0.0 %	3.5%
	ARI	7.3%	7.2%	5.11%	10.01%	2.34%	3.84%	-	0.0 %	4.3%
	Purity	50.5%	56.8%	43.49%	45.18%	49.38%	58.69%	-	58.1%	60.5%
GAN	AMI	14.8%	14%	6.7%	13.7%	7.5%	4.1%	6.0%	0 %	21.4%
	ARI	15.7%	12.6%	4.6%	11.4%	3%	5.8%	5.6%	0 %	27.9%
	Purity	58.4%	62%	42.4%	49.6%	48.9%	57.5%	46.0%	58.0%	76.5%
SimCLR	AMI	19.6%	20.1%	10.7%	13.9%	16.5%	12.5%	5.6%	6.2 %	30.2%
	ARI	16.7%	22.1%	8.6%	8.9%	11.1%	14.2%	4.5%	8.4 %	30.7%
	Purity	57.5%	68.2%	48.3%	40.9%	57.1%	67.5%	45.2%	65.1%	77.7%
DINO	AMI	1.9%	0.6%	0.3%	7.5%	0.4%	0.4%	0.3%	0.0 %	4.9%
	ARI	1.7%	0.7%	0.6%	5.5%	0.0%	0.5%	0.8%	0.5 %	6.6%
	Purity	1.9%	0.7%	0.5%	7.1%	0.4%	0.6%	41.9%	58.1%	62.9%
Volta (w/o env)	AMI	24.2%	22.8%	10.75%	19.5%	10.85%	3.6%	5.3%	0 %	35.5%
	ARI	21.7%	24%	7.58%	16.1%	6.2%	2.45%	3.7%	0 %	39.4%
	Purity	51.3%	68.3%	46.87%	54.6%	52.66%	54.7%	43.8%	58.1%	81.4%
VOLTA	AMI	25.5%	26.2%	13.8%	22.5%	17.3%	8.05%	4.2%	25.4%	50.4%
	ARI	19.3%	27.3%	8.94%	21.8%	11.4%	4.95%	6.7%	33.1%	60.3%
	Purity	63.5%	70.3%	47.7%	56.9%	57.9%	59.45%	44.8%	79.0%	88.8%

R3, Comment 3: The author did not use experiments to demonstrate why the two branch SSL structure proposed in this paper is necessary. Why don't we just use a off-the shelf SSL methods.

Response: This is a valid point and thank you for this comment. In case the reviewer is referring to the two cell augmentation branches, this structure is inspired from the architectural design of self-supervised models [Chen et al. 2020]. The main purpose of doing so is to have two visually different looking images of the exact same cells. Even though it is possible to utilize more than two branches (i.e., more than two sets of augmentations), using two branches is the simplest design and prevents the complications in the pipeline and especially the loss function. Since these augmentations are random, either one can be passed through the backbone. We have also revised the manuscript in section 2.1, page 4, lines 161-175, to elaborate more on this aspect. However, more details can be found in [LeCun et al. 2006].

That said, if the reviewer is requesting more information with regards to the necessity of the environment block in conjunction with cell block, we have added an experiment where we removed the environment branch from Volta to show the importance of the two-branch setup (cell block and environment block). The results of our experiments (Table 1, included in our

response to R3 Comment 2, above) demonstrate that the single-branch version of Volta (i.e., Volt (w/o env)) has an inferior performance, which illustrates the importance of two branch design.

R3, Comment 4: One may also imagine that the robustness and generalization capability might be challenged by this complicated pipeline.

Response: Our extensive experiments on nine datasets across six different tissue types and approximately 800,000 cells (largest in the field to our knowledge), compared to the state of the art (SOTA) baselines including off-the-shelf SSL models such as SimCLR [Chen et al. 2020] and DINO [Caron et al. 2021], demonstrate the superiority of Volta. These results confirm the robustness and generalization capability of Volta compared to currently available models.

We also believe the added design choices in the model are necessary. In the cell block, the minimal two-branch design ensures that the semantic information of the cell stays the same by gathering the embedding from different views of the same cell. On the other hand, the environment block adds the necessary information about the environment of the cell, which has been previously missing from the literature.

R3, Comment 5: the internal sample dataset, for the patch classification task, is too small to make any conclusion.

Response: Done. We have now added two new large datasets for the ovarian histotype and endometrial molecular classification tasks. More specifically, the new ovarian set is collected from 186 tissue microarray (TMA) cores (95 unique patients), and the new endometrial cohort contains 633 TMA cores (633 unique patients). Both of the new datasets include two subtypes: high-grade serous and clear cell for the ovarian cancer histotypes, and p53abn along with NSMP for the endometrial cancer cases. Our results in Supplementary Fig A5 and Supplementary Table A13 (both included below) illustrate the qualitative and quantitative improvements of our method compared to the patch-based SSL counterpart. Our results imply that the representations learned using Volta are able to cluster patients into groups that have a known corresponding histotype (ovarian set) and molecular attributes (endometrial set), which confirms the results from experiments with smaller sets.

Fig. A5 Qualitative comparison of cancer subtype clustering between VOLTA and patch-based SSL Ciga et al (2022) on ovarian dataset 2 and endometrial dataset 2

Table A13 Comparison of subtype classification for Volta and Patch-based SimCLR.

Dataset	AMI	ARI	Purity
Ovarian Datasets			
VOLTA (ours)	61.96%	72.09%	94.09%
Patch-Based SimCLR	22.53	14.68	80.65
Endometrial Datasets			
VOLTA (ours)	4.50%	5.54%	61.93%
Patch-Based SimCLR	0.99	1.66	56.76

R3, Comment 6: Another example, in Fig 2., while U-map may show some separation for different types for cells, there is no comparisons to other methods that can be used extracting the feature representations. The qualitative analysis becomes pointless without comparison.

Response: Thank you for this important comment. To address this issue, we have added UMAP visualizations of the best performing baselines such as Manual Features, Morphological Features, and SimCLR in Supplementary Figure A2 (included below), Supplementary Figure A3, and Supplementary Figure A4, respectively. The qualitative visualizations illustrated in these figures confirm the quantitative results in Table 1 (included in our response to R3 Comment 2). The superior performance of Volta is visually assessable. For instance, as shown in Supplementary Figure A2, while many overlaps can be seen in the CoNSeP, Pannuke Colon, Lizard, and MastCell datasets for the UMAP plots associated with the Manual Features method, the cell populations have more clear separation in Volta UMAPs (Figure 2).

Fig. A2 The UMAP visualization of Manual Features baseline model for different dataset

Fig. 2 Embedding space representation of each dataset using UMAP. Contours with the same color demonstrate the distribution of the learned representations by our model for that specific cell types. Despite not using labeled data in the training process, our model learns to map cells with the same type close to each other. The co-centered contours with the same color show the distribution of the representation for cells with a specific type.

R3, Comment 7: Last, the quality of writing does not meet nature standards, which lacks details on data generation, methods clarification. It reads fine for technical readers/domain readers. However, for most readers, it might be challenging to follow. Also, for many concepts/methods, citations are missing.

Response: Thank you for raising this important. Point. As suggested, we have provided more comprehensive details on data generation and methodologies and updated the manuscript accordingly (page 4 and lines 157-175, page 13 and lines 561-569, page 26 and lines 1161-

1288, page 35 and lines 1594-1628). More specifically, we have added a thorough explanation of the processes and techniques used, especially the augmentation and rationale behind developing the two-branch model, ensuring the information is complete and precise (section 2.1 and Methods of the manuscript). Furthermore, we revised the content to make it more accessible. This revision includes simplifying complex terms and ensuring that the writing is approachable for those without domain-specific knowledge, especially in data generation and preparation aspects (page 26 and lines 1161-1288). We also systematically reviewed and added necessary citations to support the concepts and methods discussed.

R3, Comment 8: Below are the listed points the author should make changes on:

10. Line 092-093: 'Furthermore, to...has to be carried out again.'

I doubt this is always the case. For cell segmentation and phenotype, model may not need to be re-trained with newly labeled datasets.

2. Line 101-103: 'The former jeopardizes ... the accuracy of the model.'

I am not sure what the author was trying to say in terms of the context. Please rephrase/clarify.

3. Line 183-186: 'For instance, PanNuke Colon and CoNSeP.'

The author should provide that information for readers who are not in cell segmentation domain.

Those acronyms can get readers totally lost. I understand that there is a section later to describe data, a brief description on the dataset and task is helpful to the reader especially those acronyms come first in the paper. Also, citation should be included.

4. Line 212-215: 'a reasonable performance ... achieved 80.2% and 76.3%.'

This statement is not that useful. The comparison does not make sense. The interpretation of reasonable performance is different from person to person. Especially the results are 62.7% and 72.6%. Those results are not reasonable to me at least. Instead, I feel you might say 'the performance achieves comparable performance when only 20% labeled data were used.'

5. In section 2.4.

Should the author perform heavier color augmentation can help on the staining variability?

6. Section 2.5.

The author spent a lot of efforts on this section while the results are constituted from a very small sample size with primarily qualitative results. I do not know if this kind of finding means anything. While we perform research discovery and this kind of finding may lead us to perform further experiments, it will not be adequate make any conclusion.

7. Line 328-334.

Refer to point 7.

8. Line 345-373.

A lot of random comments that is not related to the results and methods. The author can condense this into a few sentences for potential applications. Anyone can image any medical application.

9. Section 4.3.

The author did not describe the dataset clear enough. It is for in-domain readers. For other readers, they have no idea about the data. They do not even know what annotation, what task.

10. The author should cite and comment on this work:

'Identification of molecular cell type of breast cancer on digital histopathology images using deep learning and multiplexed fluorescence imaging.'

Response: We would like to thank the reviewer for very detailed suggestions. Please find our point-by-point responses below:

1. We agree that when a model is trained on a specific cancer type it can be applied to other samples from the same cancer type without retraining, however, our main point here is that when a model is trained on the data collected from a specific tissue type (such as ovarian) that model cannot necessary be used for other cancer types (such as breast). Therefore, the training data collection and annotation processes has to be carried out again on this new tissue type. Also, We have modified the manuscript in page 3 and lines 108-112 to clarify this.
2. Done. We have rephrased the corresponding lines to clarify this (page 3, lines 116-121 of the new manuscript).
3. Done. We have edited Section 2.1, page 4, and lines 184-186 to introduce the dataset names with the purpose to familiarize the reader with them. Additionally, we have included the citations for all of the public datasets in Supplementary Section A.1
4. Done. We have rephrased this term in page 6 and line 253-256.

5. Our experiments in Supplementary Table A5 demonstrate that the model is currently robust to staining variability. More specifically, we show that the clustering performance does not change with the stain normalization.

Table A5 Color normalization effect on unsupervised cell clustering

	w/o Color normalization			w/ Color normalization		
	AMI	ARI	Purity	AMI	ARI	Purity
VOLTA	26.1%	25.6%	70.3%	26.8%	22.9%	70.8%

6. Done. We have addressed this concern in Comment 5 of Reviewer 3.

7. We believe with including larger datasets in downstream tasks, this concern has been addressed in the revised manuscript. Please refer to response to R3 Comment 5.

8. Thank you for the insightful comment. We have modified these lines (page 9 and lines 391-402) to improve the writing.

9. Thank you for the helpful comment. We have modified the supplementary to provide more detailed description of the dataset while providing enough information for out-of-domain readers to understand the process. Please see Reviewer 1 Comment 4 for more details.

10. Thank you for the suggestion. We have cited and commented on this work in the introduction section, page 3, lines 99-104. More specifically, we have added the following statement:

“For example, in a recent study, [Han et al. 2023] developed a pipeline for segmentation and identification of several molecular features of cells from H&E images by employing supervised techniques while the ground truth data (i.e., labels) were generated through immunohistochemistry (IHC) staining and co-registration of IHC and H&E images.”

References

- Fremond, Sarah, et al. "Interpretable deep learning model to predict the molecular classification of endometrial cancer from haematoxylin and eosin-stained whole-slide images: a combined analysis of the PORTEC randomised trials and clinical cohorts." *The Lancet Digital Health* 5.2 (2023): e71-e82.
- Chen, Ting, et al. "A simple framework for contrastive learning of visual representations." *International conference on machine learning*. PMLR, 2020.
- Ciga, Ozan, Tony Xu, and Anne Louise Martel. "Self supervised contrastive learning for digital histopathology." *Machine Learning with Applications* 7 (2022): 100198.
- Hu, Bo, et al. "Unsupervised learning for cell-level visual representation in histopathology images with generative adversarial networks." *IEEE journal of biomedical and health informatics* 23.3 (2018): 1316-1328.
- Chen, Ting, et al. "A simple framework for contrastive learning of visual representations." *International conference on machine learning*. PMLR, 2020.
- Caron, Mathilde, et al. "Emerging properties in self-supervised vision transformers." *Proceedings of the IEEE/CVF international conference on computer vision*. 2021.
- Doan, Tan NN, et al. "SONNET: A self-guided ordinal regression neural network for segmentation and classification of nuclei in large-scale multi-tissue histology images." *IEEE Journal of Biomedical and Health Informatics* 26.7 (2022): 3218-3228.
- Jaume, Guillaume, et al. "Histocartography: A toolkit for graph analytics in digital pathology." *MICCAI Workshop on Computational Pathology*. PMLR, 2021.
- Ryu, Jeongun, et al. "OCELOT: Overlapped Cell on Tissue Dataset for Histopathology." *Proceedings of the IEEE/CVF Conference on Computer Vision and Pattern Recognition*. 2023.
- Han, Wenchao, et al. "Identification of molecular cell type of breast cancer on digital histopathology images using deep learning and multiplexed fluorescence imaging." *Medical Imaging 2023: Digital and Computational Pathology*. Vol. 12471. SPIE, 2023.

Reviewers' Comments:

Reviewer #1:

Remarks to the Author:

I am happy with the responses provided.

Reviewer #2:

Remarks to the Author:

All my concerns have been addressed. Thank you!

Reviewer #3:

Remarks to the Author:

The team has done great job on addressing my comments. Thus, I would recommend for publication.